# PandA: Unsupervised Learning of Parts and Appearances in the Feature Maps of GANs

**James Oldfield**[1]*, **Christos Tzelepis**[1], **Yannis Panagakis**[2]
**Mihalis A. Nicolaou**[3], **Ioannis Patras**[1]

[1]Queen Mary University of London, [2]University of Athens, [3]The Cyprus Institute

## Abstract

Recent advances in the understanding of Generative Adversarial Networks (GANs) have led to remarkable progress in visual editing and synthesis tasks, capitalizing on the rich semantics that are embedded in the latent spaces of pre-trained GANs. However, existing methods are often tailored to specific GAN architectures and are limited to either discovering global semantic directions that do not facilitate localized control, or require some form of supervision through manually provided regions or segmentation masks. In this light, we present an architecture-agnostic approach that jointly discovers factors representing spatial parts and their appearances in an entirely unsupervised fashion. These factors are obtained by applying a semi-nonnegative tensor factorization on the feature maps, which in turn enables context-aware local image editing with pixel-level control. In addition, we show that the discovered appearance factors correspond to saliency maps that localize concepts of interest, without using any labels. Experiments on a wide range of GAN architectures and datasets show that, in comparison to the state of the art, our method is far more efficient in terms of training time and, most importantly, provides much more accurate localized control. Our code is available at https://github.com/james-oldfield/PandA.

## 1 Introduction

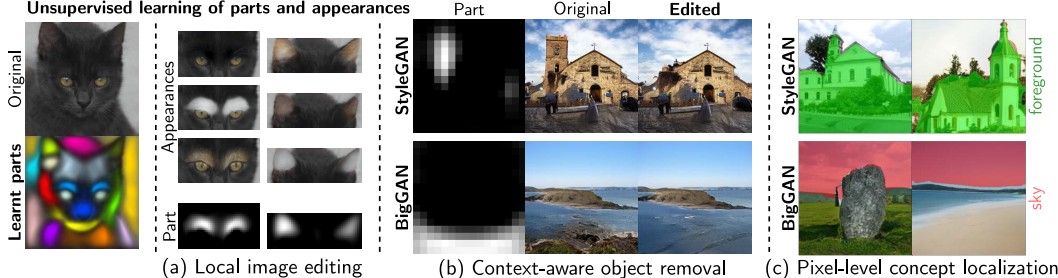

Figure 1: We propose an unsupervised method for learning a set of factors that correspond to interpretable parts and appearances in a dataset of images. These can be used for multiple tasks: (a) local image editing, (b) context-aware object removal, and (c) producing saliency maps for learnt concepts of interest.

Generative Adversarial Networks (GANs) (Goodfellow et al., 2014) constitute the state of the art (SOTA) for the task of image synthesis. However, despite the remarkable progress in this domain through improvements to the image generator's architecture (Radford et al., 2016; Karras et al., 2018; 2019; 2020b; 2021; Brock et al., 2019), their inner workings remain to a large extent unexplored. Developing a better understanding of the way in which high-level concepts are represented and composed to form synthetic images is important for a number of downstream tasks such as generative

---

*Corresponding author: j.a.oldfield@qmul.ac.uk

model interpretability (Shen et al., 2020a; Bau et al., 2019; Yang et al., 2021) and image editing (Härkönen et al., 2020; Shen & Zhou, 2021; Shen et al., 2020c; Voynov & Babenko, 2020; Tzelepis et al., 2021; Bau et al., 2020). In modern generators however, the synthetic images are produced through an increasingly complex interaction of a set of per-layer latent codes in tandem with the feature maps themselves (Karras et al., 2020b; 2019; 2021) and/or with skip connections (Brock et al., 2019). Furthermore, given the rapid pace at which new architectures are being developed, demystifying the process by which these vastly different networks model the constituent parts of an image is an ever-present challenge. Thus, many recent advances are architecture-specific (Wu et al., 2021; Collins et al., 2020; Ling et al., 2021) and a general-purpose method for analyzing and manipulating convolutional generators remains elusive.

A popular line of GAN-based image editing research concerns itself with learning so-called "interpretable directions" in the generator's latent space (Härkönen et al., 2020; Shen & Zhou, 2021; Shen et al., 2020c; Voynov & Babenko, 2020; Tzelepis et al., 2021; Yang et al., 2021; He et al., 2021; Haas et al., 2021; 2022). Once discovered, such representations of high-level concepts can be manipulated to bring about predictable changes to the images. One important question in this line of research is how latent representations are combined to form the appearance at a particular *local* region of the image. Whilst some recent methods attempt to tackle this problem (Wang et al., 2021; Wu et al., 2021; Broad et al., 2022; Zhu et al., 2021a; Zhang et al., 2021; Ling et al., 2021; Kafri et al., 2021), the current state-of-the-art methods come with a number of important drawbacks and limitations. In particular, existing techniques require prohibitively long training times (Wu et al., 2021; Zhu et al., 2021a), costly Jacobian-based optimization (Zhu et al., 2021a; 2022), and the requirement of semantic masks (Wu et al., 2021) or manually specified regions of interest (Zhu et al., 2021a; 2022). Furthermore, whilst these methods successfully find directions affecting local changes, optimization must be performed on a per-region basis, and the resulting directions do not provide *pixel-level* control–a term introduced by Zhu et al. (2021a) referring to the ability to *precisely* target specific pixels in the image.

In this light, we present a fast unsupervised method for *jointly* learning factors for interpretable parts and their appearances (we thus refer to our method as *PandA*) in pre-trained convolutional generators. Our method allows one to both interpret and edit an image's style at discovered local semantic regions of interest, using the learnt appearance representations. We achieve this by formulating a constrained optimization problem with a semi-nonnegative tensor decomposition of the dataset of deep feature maps $\mathcal{Z} \in \mathbb{R}^{M \times H \times W \times C}$ in a convolutional generator. This allows one to accomplish a number of useful tasks, prominent examples of which are shown in Fig. 1. Firstly, our learnt representations of appearance across samples can be used for the popular task of local image editing (Zhu et al., 2021a; Wu et al., 2021) (for example, to change the colour or texture of a cat's ears as shown in Fig. 1 (a)). Whilst the state-of-the-art methods (Zhu et al., 2021a; Wu et al., 2021; Zhu et al., 2022) provide fine-grained control over a target region, they adopt an "annotation-first" approach, requiring an end-user to first manually specify a ROI. By contrast, our method fully exploits the unsupervised learning paradigm, wherein such concepts are discovered automatically and without any manual annotation. These discovered semantic regions can then be chosen, combined, or even modified by an end-user as desired for local image editing.

More interestingly still, through a generic decomposition of the feature maps our method identifies representations of common concepts (such as "background") in *all* generator architectures considered (all 3 StyleGANs (Karras et al., 2019; 2020b; 2021), ProgressiveGAN (Karras et al., 2018), and BigGAN (Brock et al., 2019)). This is a surprising finding, given that these generators are radically different in architecture. By then editing the feature maps using these appearance factors, we can thus, for example, *remove* specific objects in the foreground (Fig. 1 (b)) in all generators, seamlessly replacing the pixels at the target region with the background appropriate to each image.

However, our method is useful not only for local image editing, but also provides a straightforward way to localize the learnt appearance concepts in the images. By expressing activations in terms of our learnt appearance basis, we are provided with a *visualization* of how much of each of the appearance concepts are present at each spatial location (i.e., saliency maps for concepts of interest). By then thresholding the values in these saliency maps (as shown in Fig. 1 (c)), we can localize the learnt appearance concepts (such as sky, floor, or background) in the images–without the need for supervision at any stage.

We show exhaustive experiments on 5 different architectures (Karras et al., 2020b; 2018; 2021; 2019; Brock et al., 2019) and 5 datasets (Deng et al., 2009; Choi et al., 2020; Karras et al., 2019; Yu et al.,

Table 1: A high-level comparison of our method to the SOTA for local image editing. "Training time" denotes the total training time required to produce the images for the quantitative comparisons.

| | Manual ROI–free | Semantic mask–free | Pixel-level control | Architecture-agnostic | Style diversity | Training time (mins) |
|---|---|---|---|---|---|---|
| StyleSpace (Wu et al., 2021) | ✓ | ✗ | ✗ | ✗ | ✓ | 177.12 |
| LowRankGAN (Zhu et al., 2021a) | ✗ | ✓ | ✗ | ✓ | ✗ | 324.21 |
| ReSeFa (Zhu et al., 2022) | ✗ | ✓ | ✗ | ✓ | ✗ | 347.79 |
| **Ours** | ✓ | ✓ | ✓ | ✓ | ✓ | **0.87** |

2015; Karras et al., 2020a). Our method is not only orders of magnitude faster than the SOTA, but also showcases superior performance at the task of local image editing, both qualitatively and quantitatively. Our contributions can be summarized as follows:

- We present an architecture-agnostic unsupervised framework for learning factors for both the parts and the appearances of images in pre-trained GANs, that enables local image editing. In contrast to the SOTA (Zhu et al., 2021a; Wu et al., 2021), our approach requires neither semantic masks nor manually specified ROIs, yet offers more precise pixel-level control.

- Through a semi-nonnegative tensor decomposition of the generator's feature maps, we show how one can learn sparse representations of semantic parts of images by formulating and solving an appropriate constrained optimization problem.

- We show that the proposed method learns appearance factors that correspond to semantic concepts (e.g., background, sky, skin), which can be localized in the image through saliency maps.

- A rigorous set of experiments show that the proposed approach allows for more accurate local image editing than the SOTA, while taking only a fraction of the time to train.

## 2 RELATED WORK

Generative Adversarial Networks (GANs) (Goodfellow et al., 2014) continue to push forward the state of the art for the task of image synthesis through architectural advances such as the use of convolutions (Radford et al., 2016), progressive growing (Karras et al., 2018), and style-based architectures (Karras et al., 2019; 2020b; 2021). Understanding the representations induced by these networks for interpretation (Bau et al., 2019; Shen et al., 2020a; Yang et al., 2021) and control (Shen & Zhou, 2021; Härkönen et al., 2020; Voynov & Babenko, 2020; Georgopoulos et al., 2021; Tzelepis et al., 2021; Zhu et al., 2021a; Bounareli et al., 2022; Wu et al., 2021; Abdal et al., 2021) has subsequently received much attention.

However, whilst several methods identify ways of manipulating the latent space of GANs to bring about global semantic changes–either in a supervised (Goetschalckx et al., 2019; Plumerault et al., 2020; Shen et al., 2020c;a) or unsupervised (Voynov & Babenko, 2020; Shen & Zhou, 2021; Härkönen et al., 2020; Tzelepis et al., 2021; Oldfield et al., 2021) manner–many of them struggle to apply *local* changes to regions of interest in the image. In this framework of local image editing, one can swap certain parts between images (Collins et al., 2020; Jakoel et al., 2022; Chong et al., 2021; Suzuki et al., 2018; Kim et al., 2021; Bau et al., 2020), or modify the style at particular regions (Wang et al., 2021; Wu et al., 2021; Broad et al., 2022; Zhu et al., 2021a; Zhang et al., 2021; Ling et al., 2021; Kafri et al., 2021). This is achieved with techniques such as clustering (Collins et al., 2020; Zhang et al., 2021; Broad et al., 2022; Kafri et al., 2021), manipulating the AdaIN (Huang & Belongie, 2017) parameters (Wu et al., 2021; Wang et al., 2021), or/and operating on the feature maps themselves (Wang et al., 2021; Broad et al., 2022; Zhang et al., 2021) to aid the locality of the edit. Other approaches employ additional latent spaces or architectures (Kim et al., 2021; Ling et al., 2021), require the computation of expensive gradient maps (Wang et al., 2021; Wu et al., 2021) and semantic segmentation masks/networks (Wu et al., 2021; Zhu et al., 2021b; Ling et al., 2021), or require manually specified regions of interest (Zhu et al., 2021a; 2022). In contrast to related work, our method automatically learns both the parts and a diverse set of global appearances, in a fast unsupervised procedure without any semantic masks. Additionally, our method allows for *pixel-level* control (Zhu et al., 2021a). For example, one can choose to modify a single eye only in a face, which is not possible with the SOTA (Zhu et al., 2021a; 2022). Our method and its relationship to the SOTA for local image editing is summarized in Table 1.

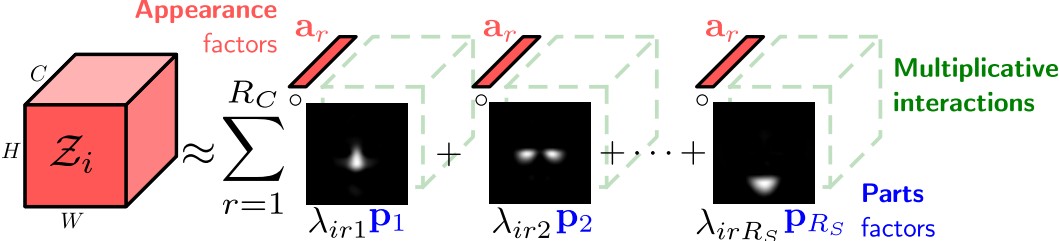

Figure 2: An overview of our method. We decompose a dataset of generator's activations $\mathcal{Z}_i \in \mathbb{R}^{\tilde{H} \times W \times C}$ with a separable model. Each factor has an intuitive interpretation: the factors for the spatial modes $\mathbf{p}_j$ control the parts, determining at *which spatial locations* in the feature maps the various appearances $\mathbf{a}_k$ are present, through their multiplicative interactions.

From a methodological standpoint, most closely related to our method are the works of Collins et al. (2020; 2018). Both of these perform clustering in the activation space for parts-based representations in generators (Collins et al., 2020) and CNNs (Collins et al., 2018) respectively. However, Collins et al. (2018) considers only discriminative networks for locating common semantic regions in CNNs, whilst we additionally focus on image editing tasks in GANs. On the other hand, Collins et al. (2020) does not jointly learn representations of *appearances*. Therefore Collins et al. (2020) is limited to swapping parts between two images, and is additionally StyleGAN-specific, unlike our method that offers a generic treatment of convolutional generators.

## 3 METHODOLOGY

In this section, we detail our approach for jointly learning interpretable parts and their appearances in pre-trained GANs, in an unsupervised manner. We begin by establishing the notation used throughout the paper in Section 3.1. We then introduce our proposed separable model in Section 3.2, and our optimization objective in Section 3.3. In Section 3.4 we describe our initialization strategies, and finally in Section 3.5 we describe how to refine the learnt global parts factors.

### 3.1 NOTATION

We use uppercase (lowercase) boldface letters to refer to matrices (vectors), e.g., $\mathbf{X}$ ($\mathbf{x}$), and calligraphic letters for higher-order tensors, e.g., $\mathcal{X}$. We refer to each element of an $N^{\text{th}}$ order tensor $\mathcal{X} \in \mathbb{R}^{I_1 \times I_2 \times \cdots \times I_N}$ using $N$ indices, i.e., $\mathcal{X}(i_1, i_2, \ldots, i_N) \triangleq x_{i_1 i_2 \ldots i_N} \in \mathbb{R}$. The **mode-$n$ fibers** of a tensor are the column vectors formed when fixing all but the $n^{\text{th}}$ mode's indices (e.g., $\mathbf{x}_{:jk} \in \mathbb{R}^{I_1}$ are the mode-1 fibers). A tensor's mode-$n$ fibers can be stacked along the columns of a matrix, giving us the **mode-$n$ unfolding** denoted as $\mathbf{X}_{(n)} \in \mathbb{R}^{I_n \times \bar{I}_n}$ with $\bar{I}_n = \prod_{\substack{t=1 \\ t \neq n}}^{N} I_t$ (Kolda & Bader, 2009). We denote a pre-trained convolutional GAN generator with $G$, and use $G_{[l:]}$ to refer to the partial application of the last $l$ layers of the generator only.

### 3.2 A SEPARABLE MODEL OF PARTS AND APPEARANCES

A convolutional generator maps each latent code $\mathbf{z}_i \sim \mathcal{N}(\mathbf{0}, \mathbf{I})$ to a synthetic image $\mathcal{X}_i \in \mathbb{R}^{\tilde{H} \times \tilde{W} \times \tilde{C}}$ via a sequence of 2D transpose convolutions. The intermediate convolutional features $\mathcal{Z}_i \in \mathbb{R}^{H \times W \times C}$ at each layer have a very particular relationship to the output image. Concretely, each spatial activation (Olah et al., 2018) (which can be thought of as a spatial coordinate in the feature maps in Fig. 2 indexed with an $(h, w)$ tuple) affects a specific patch in the output image (Collins et al., 2020). At each of these spatial positions, a channel fiber $\mathbf{z}_{ihw:} \in \mathbb{R}^C$ lies depth-wise along the activation tensor, determining its value. With this understanding, we propose to factor the spatial and channel modes *separately* with a tensor decomposition, providing an intuitive separation into representations of the images' parts and appearances. This provides a simple interface for local image editing. We suggest that representations of a set of interpretable parts for local image editing should have two properties:

1. *Non-negativity*: the representations ought to be *additive* in nature, thus corresponding to semantic parts of the images (Lee & Seung, 1999).

2. *Sparsity*: the parts should span disjoint spatial regions, capturing different localized patterns in space, as opposed to global ones (Yang & Oja, 2010; Yuan et al., 2009).

Concretely, given the dataset's intermediate feature maps $\mathcal{Z} \in \mathbb{R}^{N \times H \times W \times C}$ from the pre-trained generator, each sample's mode-3 unfolding $\mathbf{Z}_{i(3)} \in \mathbb{R}^{C \times S}$ contains in its columns the channel-wise activations at each of the $S \triangleq H \cdot W$ spatial positions in the feature maps.[1] We propose a separable factorization of the form

$$\mathbf{Z}_{i(3)} = \mathbf{A}\mathbf{\Lambda}_i\mathbf{P}^\top \tag{1}$$

$$= \underbrace{\begin{bmatrix} | & & | \\ \mathbf{a}_1 & \cdots & \mathbf{a}_{R_C} \\ | & & | \end{bmatrix}}_{\text{Appearance}} \underbrace{\begin{bmatrix} \lambda_{i11} & \lambda_{i12} & \cdots \\ \vdots & \ddots & \\ \lambda_{iR_C1} & \cdots & \lambda_{iR_CR_S} \end{bmatrix}}_{\text{Sample } i\text{'s coefficients}} \underbrace{\begin{bmatrix} - & \mathbf{p}_1^\top & - \\ & \vdots & \\ - & \mathbf{p}_{R_S}^\top & - \end{bmatrix}}_{\text{Parts}}, \tag{2}$$

where $\mathbf{A} \in \mathbb{R}^{C \times R_C}$ are the global appearance factors and $\mathbf{P} \geq \mathbf{0} \in \mathbb{R}^{S \times R_S}$ are the global parts factors (with $R_C \leq C, R_S \leq S$), jointly learnt across many samples in a dataset. Intuitively, the coefficients $\lambda_{ijk}$ encode how much of appearance $\mathbf{a}_j$ is present at part $\mathbf{p}_k$ in sample $i$'s feature maps $\mathbf{Z}_{i(3)}$. We show our proposed separable decomposition schematically in Fig. 2. Each *non-negative* parts factor $\mathbf{p}_k \in \mathbb{R}^S \geq \mathbf{0}$ spans a spatial sub-region of the feature maps, corresponding to a semantic part. The various appearances and textures present throughout the dataset are encoded in the appearance factors $\mathbf{a}_j \in \mathbb{R}^C$ and lie along the depth-wise channel mode of the feature maps. This formulation facilitates modelling the multiplicative interactions (Jayakumar et al., 2020) between the parts and appearance factors. Concretely, due to the outer product, the factors relating to the parts control the spatial regions at which the various appearance factors are present. The parts factors thus function similarly to semantic masks, but rather are learnt jointly and in an entirely unsupervised manner. This is particularly useful for datasets for which segmentation masks are not readily available.

## 3.3 OBJECTIVE

We propose to solve a constrained optimization problem that leads to the two desirable properties outlined in Section 3.2. We impose hard non-negativity constraints on the parts factors $\mathbf{P}$ to achieve property 1, and encourage both factor matrices to be column-orthonormal for property 2 (which has been shown to lead to sparser representations (Ding et al., 2006; Yang & Laaksonen, 2007; Yang & Oja, 2010; Yuan et al., 2009), and has intricate connections to clustering (Ding et al., 2005; Li & Ding, 2006)). We achieve this by formulating a single reconstruction objective as follows. Let $\mathcal{Z} \in \mathbb{R}^{N \times C \times S}$ be a batch of $N$ samples' mode-3 unfolded intermediate activations. Then our constrained optimization problem is

$$\min_{\mathbf{A},\mathbf{P}} \mathcal{L}(\mathcal{Z}, \mathbf{A}, \mathbf{P}) = \min_{\mathbf{A},\mathbf{P}} \sum_{i=1}^N ||\mathbf{Z}_i - \mathbf{A}\left(\mathbf{A}^\top \mathbf{Z}_i \mathbf{P}\right) \mathbf{P}^\top||_F^2 \quad \text{s.t. } \mathbf{P} \geq \mathbf{0}. \tag{3}$$

A good reconstruction naturally leads to orthogonal factor matrices (e.g., $\mathbf{P}^\top\mathbf{P} \approx \mathbf{I}_{R_S}$ for $\mathbf{P} \in \mathbb{R}^{S \times R_S}$ with $S \geq R_S$) without the need for additional hard constraints (Le et al., 2013). What's more, each parts factor (column of $\mathbf{P}$) is encouraged to span a distinct spatial region to simultaneously satisfy both the non-negativity and orthonormality-via-reconstruction constraints. However, this problem is non-convex. We thus propose to break the problem into two sub-problems in $\mathbf{A}$ and $\mathbf{P}$ separately, applying a form of block-coordinate descent (Lin, 2007), optimizing each factor matrix separately whilst keeping the other fixed. The gradients of the objective function in Eq. (3) with respect to the two factor matrices (see the supplementary material for the derivation) are given by

$$\nabla_{\mathbf{P}}\mathcal{L} = 2\left(\sum_{i=1}^N \bar{\mathbf{P}}\mathbf{Z}_i^\top \bar{\mathbf{A}}\bar{\mathbf{A}}\mathbf{Z}_i\mathbf{P} + \mathbf{Z}_i^\top \bar{\mathbf{A}}\bar{\mathbf{A}}\mathbf{Z}_i\bar{\mathbf{P}}\mathbf{P} - 2\mathbf{Z}_i^\top \bar{\mathbf{A}}\mathbf{Z}_i\mathbf{P}\right), \tag{4}$$

$$\nabla_{\mathbf{A}}\mathcal{L} = 2\left(\sum_{i=1}^N \bar{\mathbf{A}}\mathbf{Z}_i\bar{\mathbf{P}}\bar{\mathbf{P}}\mathbf{Z}_i^\top \mathbf{A} + \mathbf{Z}_i\bar{\mathbf{P}}\bar{\mathbf{P}}\mathbf{Z}_i^\top \bar{\mathbf{A}}\mathbf{A} - 2\mathbf{Z}_i\bar{\mathbf{P}}\mathbf{Z}_i^\top \mathbf{A}\right), \tag{5}$$

---

[1]Intuitively, $\mathbf{Z}_{i(3)} \in \mathbb{R}^{C \times S}$ can be viewed simply as a 'reshaping' of the $i^{\text{th}}$ sample's feature maps that combines the height and width modes into a single $S$-dimensional 'spatial' mode.

with $\bar{\mathbf{P}} \triangleq \mathbf{P}\mathbf{P}^\top$ and $\bar{\mathbf{A}} \triangleq \mathbf{A}\mathbf{A}^\top$. After a gradient update for the parts factors $\mathbf{P}$, we project them onto the non-negative orthant (Lin, 2007) with $\max\{\mathbf{0}, \cdot\}$. This leads to our alternating optimization algorithm, outlined in Algorithm 1.

---

**Algorithm 1:** Block-coordinate descent solution to Eq. (3)

---

**Input** : $\mathcal{Z} \in \mathbb{R}^{M \times C \times S}$ ($M$ lots of mode-3-unfolded activations), $R_C, R_S \in \mathbb{R}$ (ranks), $\lambda \in \mathbb{R}$ (learning rate), and $T$ (# iterations).

**Output** : $\mathbf{P} \in \mathbb{R}^{S \times R_S}, \mathbf{A} \in \mathbb{R}^{C \times R_C}$ (*parts* and *appearance* factors).

**Initialise**

$\quad \mathbf{U}, \mathbf{\Sigma}, \mathbf{V}^\top \leftarrow \text{SVD}\left(\mathbf{Z}_{(2)}\mathbf{Z}_{(2)}^\top\right);$

$\quad \mathbf{A}^{(1)} \leftarrow \mathbf{U}_{:R_C};$

$\quad \mathbf{P}^{(1)} \sim \mathcal{U}(0, 0.01);$

**for** $t = 1$ **to** $T$ **do**

$\quad \mathbf{P}^{(t+1)} \leftarrow \max\left\{\mathbf{0}, \mathbf{P}^{(t)} - \lambda \cdot \nabla_{\mathbf{P}^{(t)}}\mathcal{L}\left(\mathcal{Z}, \mathbf{A}^{(t)}, \mathbf{P}^{(t)}\right)\right\};$      `// PGD step`

$\quad \mathbf{A}^{(t+1)} \leftarrow \mathbf{A}^{(t)} - \lambda \cdot \nabla_{\mathbf{A}^{(t)}}\mathcal{L}\left(\mathcal{Z}, \mathbf{A}^{(t)}, \mathbf{P}^{(t+1)}\right);$

**end**

---

Upon convergence of Algorithm 1, to modify an image $i$ at region $k$ with the $j^{\text{th}}$ appearance with desired magnitude $\alpha \in \mathbb{R}$, we compute the forward pass from layer $l$ onwards in the generator with $\mathcal{X}'_i = G_{[l:]}\left(\mathbf{Z}_i + \alpha \mathbf{a}_j \hat{\mathbf{p}}_k^\top\right)$, with $\hat{\mathbf{p}}_k$ being the normalized parts factor of interest.

## 3.4 INITIALIZATION

Let $\mathcal{Z} \in \mathbb{R}^{N \times C \times S}$ be a batch of $N$ mode-3 unfolded feature maps as in Section 3.3. A common initialization strategy (Cichocki et al., 2009; Boutsidis & Gallopoulos, 2008; Yuan et al., 2009) for non-negative matrix/tensor decompositions is via a form of HOSVD (Tucker, 1966; Lu et al., 2008). Without non-negativity constraints, the channel factor matrix subproblem has a closely related closed-form solution given by the first $R_C$ left-singular vectors of the mode-2 unfolding of the activations expressed in terms of the parts basis (proof given in Appendix B of Xu et al. (2005)). We thus initialize the channel factors at time-step $t = 1$ with $\mathbf{A}^{(1)} \triangleq \mathbf{U}_{:R_C}$ where $\mathbf{U}_{:R_C}$ are the first $R_C$-many left-singular vectors of $\mathbf{Z}_{(2)}\mathbf{Z}_{(2)}^\top$. Later on in Section 4.1.2 we demonstrate the benefits of this choice, including its usefulness for locating interpretable appearances.

## 3.5 PARTS FACTORS REFINEMENT

The formulation in Eq. (3) for learning parts and appearances makes the implicit assumption that the samples are spatially aligned. However, this does not always hold in practice, and therefore the global parts are not always immediately useful for datasets with no alignment. To alleviate this requirement, we propose a fast optional "refinement" step of the global parts factors $\mathbf{P} \in \mathbb{R}^{S \times R_s}$ from Eq. (3) to specialize them to sample-specific parts factors $\tilde{\mathbf{P}}_i \in \mathbb{R}^{S \times R_s}$ for sample $i$. Given the $i^{\text{th}}$ target sample's intermediate activations $\mathbf{Z}_i \in \mathbb{R}^{C \times S}$, we optimize a few steps of a similar constrained optimization problem as before:

$$\min_{\tilde{\mathbf{P}}_i} \mathcal{L}_R(\mathbf{Z}_i, \mathbf{A}, \tilde{\mathbf{P}}_i) = \min_{\tilde{\mathbf{P}}_i} ||\mathbf{Z}_i - \mathbf{A}\left(\mathbf{A}^\top \mathbf{Z}_i \tilde{\mathbf{P}}_i\right)\tilde{\mathbf{P}}_i^\top||_F^2 \quad \text{s.t. } \tilde{\mathbf{P}}_i \geq \mathbf{0}. \tag{6}$$

We analyze in Appendix B.0.2 the benefits of this refinement step, and compare the global parts factors to the refined factors.

## 4 EXPERIMENTS

In this section we present a series of experiments to validate the method and explore its properties. We begin in Section 4.1 by focusing on using the method for interpretation: showing how one can generate saliency maps for concepts of interest and remove the foreground at target locations.

Following this, we showcase local image editing results on 5 GANs in Section 4.2. In Appendix B we present ablation studies to further justify and motivate our method.

## 4.1 INTERPRETING THE APPEARANCE VECTORS

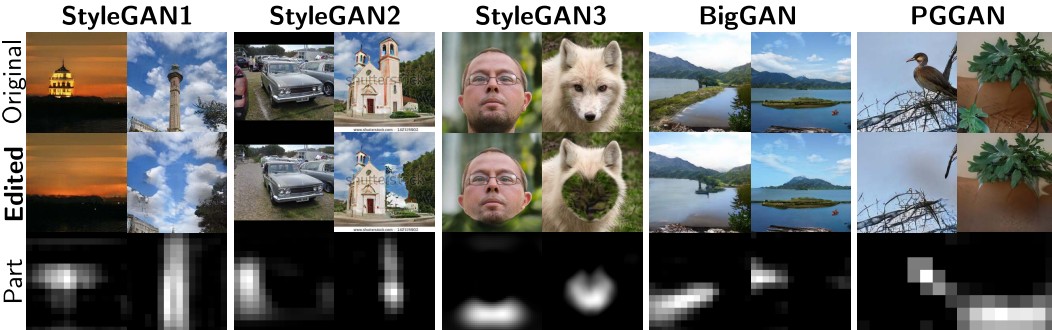

Figure 3: Our architecture-agnostic method discovers a representation of the "background" concept in the feature maps, which allows us to remove objects in a context-aware manner for all 5 generators.

Using the learnt appearance basis $\mathbf{A}$, one can straightforwardly visualize "how much" of each column is present at each spatial location via a change of basis. In particular, the element at row $c$ and column $s$ of the activations expressed in terms of the appearance basis $\mathbf{A}^\top \mathbf{Z}_i \in \mathbb{R}^{R_C \times S}$ encodes how much of appearance $c$ is present at spatial location $s$, for a particular sample $i$ of interest. This transformation provides a visual understanding of the concepts controlled by the columns by observing the semantic regions in the image at which these values are the highest.

### 4.1.1 GENERIC CONCEPTS SHARED ACROSS GAN ARCHITECTURES

The analysis above leads us to make an interesting discovery. We find that our model frequently learns an appearance vector for a high-level "background" concept in *all* 5 generator architectures. This is a surprising finding–one would not necessarily expect these radically different architectures to encode concepts in the same manner (given that many existing methods are architecture-specific), let alone that they could be extracted with a single unsupervised approach. We can thus use this learnt "background" appearance vector to remove objects in a context-aware manner, as shown on all 5 generators and numerous datasets in Fig. 3.

### 4.1.2 VISUALIZING AND LOCALIZING APPEARANCE VECTORS

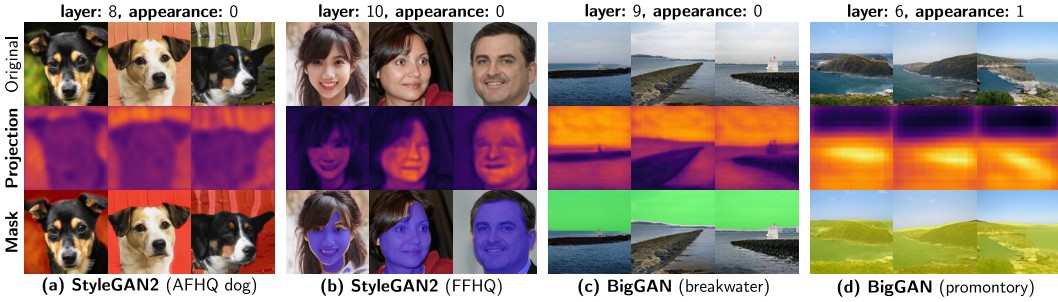

Figure 4: Visualizing the coordinates in the appearance basis ($2^{\text{nd}}$ row), one can interpret how much of each appearance vector is present at each spatial patch. For example, we see appearance vectors at various layers very clearly corresponding to (a) background, (b) skin, (c) sky, and (d) foreground.

Through the change of basis $\mathbf{A}^\top \mathbf{Z}_i$ we can identify the pixels in the image that are composed of the concept $k$ of interest (e.g., the "background" concept), offering an interpretation of the images' semantic content. We first compute the saliency map $\mathbf{m}_{ik} = \mathbf{a}_k^\top \mathbf{Z}_i \in \mathbb{R}^S$, whose elements encode the magnitude of concept $k$ at each spatial position in the $i^{\text{th}}$ sample. This can be reshaped into a

square matrix and visualized as an image to localize the $k^{\text{th}}$ concept in the image, as shown in row 2 of Fig. 4. We then additionally perform a simple binary classification following Voynov & Babenko (2020). We classify each pixel $j$ as an instance of concept $k$ or not with $\tilde{m}_{ikj} = [m_{ikj} \geq \mu_k]$, where $\mu_k = \frac{1}{N \cdot S} \sum_{n,s} m_{nks} \in \mathbb{R}$ is the mean magnitude of the $k^{\text{th}}$ concept in $N$ samples. We show this in row 3 of Fig. 4 for various datasets and GANs. For example, this analysis allows us to identify (and localize) appearance vectors in various generators that control concepts including "foreground", "skin", and "sky", shown in Fig. 4 (b-d). We find this visualization to be most useful for understanding the first few columns of $\mathbf{A}$, which control the more prominent high-level visual concepts in the dataset due to our SVD-based initialization outlined in Section 3.4.

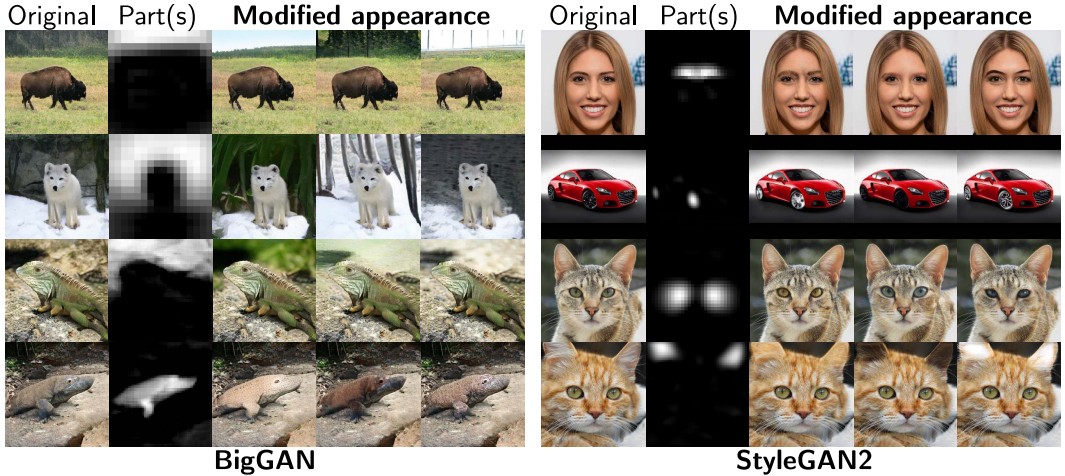

Figure 5: Local image editing on a number of architectures and datasets, using both the global and refined parts factors. At each column, the original image is edited at the target part with a different appearance vector (many more examples are shown in Appendix C.3).

## 4.2 LOCAL IMAGE EDITING

Next, we showcase our method's ability to perform local image editing in pre-trained GANs, on 5 generators and 5 datasets (ImageNet (Deng et al., 2009), AFHQ (Choi et al., 2020), FFHQ (Karras et al., 2019), LSUN (Yu et al., 2015), and MetFaces (Karras et al., 2020a)). In Fig. 5 we show a number of interesting local edits achievable with our method, using both the global and refined parts factors. Whilst we can manipulate the style at common regions such as the eyes with the global parts factors, the refined parts factors allow one to target regions such as an individual's clothes, or their background. Many more examples are shown in Appendix C.3. One is not limited to this set of learnt parts however. For example, one can draw a ROI by hand at test-time or modify an existing part–an example of this is shown in Appendix C.1. This way, pixel-level control (e.g., opening only a single eye of a face) is achievable in a way that is not possible with the SOTA methods (Zhu et al., 2021a; Wu et al., 2021).

We next compare our method to state-of-the-art GAN-based image editing techniques in Fig. 6. In particular, we train our model at layer 5 using $R_S = 8$ global parts factors, with no refinement. As can be seen, SOTA methods such as LowRank-GAN (Zhu et al., 2021a) excel at enlarging the eyes in a photo-realistic manner. However, we frequently find the surrounding regions to change as well. This is seen clearly by visualizing the mean squared error (Collins et al., 2020) between the original images and their edited counterparts, shown in every second row of Fig. 6. We further quantify this ability to affect local edits in the section that follows.

### 4.2.1 QUANTITATIVE RESULTS

We compute the ratio of the distance between the pixels of the original and edited images in the region of 'disinterest', over the same quantity with the region of interest:

$$\text{ROIR}(\mathcal{M}, \mathcal{X}, \mathcal{X}') = \frac{1}{N} \sum_{i=1}^{N} \frac{||\,(\mathbf{1} - \mathcal{M}) * (\mathcal{X}_i - \mathcal{X}'_i)\,||}{||\mathcal{M} * (\mathcal{X}_i - \mathcal{X}'_i)\,||}, \tag{7}$$

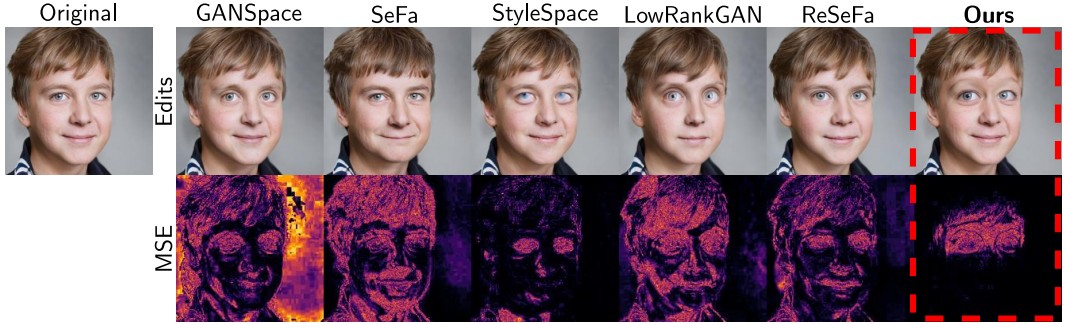

Figure 6: Qualitative comparison to SOTA methods editing the "eyes" ROI. We also show the mean squared error (Collins et al., 2020) between the original images and their edited counterparts, highlighting the regions that change.

Table 2: ROIR (↓) of Eq. (7) for 10k FFHQ samples per local edit.

| | Eyes | Nose | Open mouth | Smile |
|---|---|---|---|---|
| GANSpace (Härkönen et al., 2020) | 2.80±1.22 | 4.89±2.11 | 3.25±1.33 | 2.44±0.89 |
| SeFa (Shen & Zhou, 2021) | 5.01±1.90 | 6.89±3.04 | 3.45±1.12 | 5.04±2.22 |
| StyleSpace (Wu et al., 2021) | 1.26±0.70 | 1.70±0.82 | 1.24±0.44 | 2.06±1.62 |
| LowRankGAN (Zhu et al., 2021a) | 1.78±0.59 | 5.07±2.06 | 1.82±0.60 | 2.31±0.76 |
| ReSeFa (Zhu et al., 2022) | 2.21±0.85 | 2.92±1.29 | 1.69±0.65 | 1.87±0.75 |
| **Ours** | **1.04±0.33** | **1.17±0.44** | **1.04±0.39** | **1.05±0.38** |

where $\mathcal{M} \in [0,1]^{H \times W \times C}$ is an $H \times W$ spatial mask (replicated along the channel mode) specifying the region of interest, $\mathbf{1}$ is a 1-tensor, and $\mathcal{X}, \mathcal{X}' \in \mathbb{R}^{N \times \tilde{H} \times \tilde{W} \times \tilde{C}}$ are the batch of original and edited versions of the images respectively. A small ROIR indicates more 'local' edits, through desirable change to the ROI (large denominator) and little change elsewhere (small numerator). We compute this metric for our method and SOTA baselines in Table 2, for a number of regions of interest. As can be seen, our method consistently produces more local edits than the SOTA for a variety of regions of interest. We posit that the reason for this is due to our operating directly on the feature maps, where the spatial activations have a direct relationship to a patch in the output image. Many more comparisons and results can be found in Appendix C.

## 5 CONCLUSION

In this paper, we have presented a fast unsupervised algorithm for learning interpretable parts and their appearances in pre-trained GANs. We have shown experimentally how our method outperforms the state of the art at the task of local image editing, in addition to being orders of magnitude faster to train. We showed how one can identify and manipulate generic concepts in 5 generator architectures. We also believe that our method's ability to visualize the learnt appearance concepts through saliency maps could be a useful tool for network interpretability.

**Limitations** Whilst we have demonstrated that our method can lead to more precise control, the approach is not without its limitations. Such *strictly* local editing means that after modifying a precise image region, any expected influence on the rest of the image is not automatically accounted for. As a concrete example, one can remove trees from an image, but any shadow they may have cast elsewhere is not also removed automatically. Additionally, we find that methods editing the feature maps have a greater tendency to introduce artifacts relative to methods working on the latent codes. This is one potential risk with the freedom of pixel-level control–adding appearance vectors at arbitrary spatial locations does not always lead to photorealistic edits. We hope to address this in future work.

**Acknowledgments** This work was supported by the EU H2020 AI4Media No. 951911 project.

# 6 REPRODUCIBILITY STATEMENT

Efforts have been made throughout to ensure the results are reproducible, and the method easy to implement. We provide *full* source code in the supplementary material folder. This contains jupyter notebooks to reproduce the concept localizations from Section 4.1.2, the qualitative results from Section 4.2, and a full demo including the training and refinement objective of Section 3.5 for easy training and editing with new generators. We also provide pre-trained models that can be downloaded by following the link in the supplementary material's `readme.md`.

# 7 ETHICS STATEMENT

The development of any new method such as PandA facilitating the ability to edit images brings a certain level of risk. For example, using the model, bad actors may more easily edit images for malicious ends or to spread misinformation. Additionally, it's important to highlight that through our use of pre-trained models we inherit any bias present in the generators or the datasets on which they were trained. Despite these concerns, we believe our method leads to more interpretable and transparent image synthesis–for example, through our concept localization one has a much richer understanding of which attributes appear in the generated images, and where.

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

# A  GRADIENTS

We first provide a derivation of the gradients of the main paper's loss function with respect to each factor. The main paper's objective function is given by

$$\mathcal{L} = \sum_{i=1}^{N} ||\mathbf{Z}_i - \mathbf{A}(\mathbf{A}^\top \mathbf{Z}_i \mathbf{P})\mathbf{P}^\top||_F^2$$

$$= \sum_{i=1}^{N} \text{tr}\left( \left(\mathbf{Z}_i - \mathbf{A}(\mathbf{A}^\top \mathbf{Z}_i \mathbf{P})\mathbf{P}^\top\right)^\top \left(\mathbf{Z}_i - \mathbf{A}(\mathbf{A}^\top \mathbf{Z}_i \mathbf{P})\mathbf{P}^\top\right) \right)$$

$$= \sum_{i=1}^{N} \underbrace{\text{tr}\left(\mathbf{Z}_i^\top \mathbf{Z}_i\right)}_{\mathcal{L}_a} - 2\underbrace{\text{tr}\left(\mathbf{P}\mathbf{P}^\top \mathbf{Z}_i^\top \mathbf{A}\mathbf{A}^\top \mathbf{Z}_i\right)}_{\mathcal{L}_b} + \underbrace{\text{tr}\left(\mathbf{P}\mathbf{P}^\top \mathbf{Z}_i^\top \mathbf{A}\mathbf{A}^\top \mathbf{A}\mathbf{A}^\top \mathbf{Z}_i \mathbf{P}\mathbf{P}^\top\right)}_{\mathcal{L}_c}.$$

Clearly, $\nabla_\mathbf{P} \mathcal{L}_a = \mathbf{0}$. The term $\nabla_\mathbf{P} \mathcal{L}_b$ is of the form $\frac{\partial}{\partial \mathbf{P}} \text{tr}\left(\mathbf{P}\mathbf{P}^\top \mathbf{X}\right)$. Thus we have

$$\nabla_\mathbf{P} \mathcal{L}_b = \left(\mathbf{Z}_i^\top \mathbf{A}\mathbf{A}^\top \mathbf{Z}_i + \mathbf{Z}_i^\top \mathbf{A}\mathbf{A}^\top \mathbf{Z}_i\right)\mathbf{P} = 2 \cdot \mathbf{Z}_i^\top \mathbf{A}\mathbf{A}^\top \mathbf{Z}_i \mathbf{P}. \tag{8}$$

The final term $\nabla_\mathbf{P} \mathcal{L}_c$ has the form $\frac{\partial}{\partial \mathbf{P}} \text{tr}\left(\mathbf{P}\mathbf{P}^\top \mathbf{X}\mathbf{P}\mathbf{P}^\top\right)$. Through the chain rule, we have

$$\nabla_\mathbf{P} \mathcal{L}_c = 2\left(\mathbf{P}\mathbf{P}^\top \left(\mathbf{Z}_i^\top \mathbf{A}\mathbf{A}^\top \mathbf{A}\mathbf{A}^\top \mathbf{Z}_i\right)\mathbf{P} + \left(\mathbf{Z}_i^\top \mathbf{A}\mathbf{A}^\top \mathbf{A}\mathbf{A}^\top \mathbf{Z}_i\right)\mathbf{P}\mathbf{P}^\top \mathbf{P}\right). \tag{9}$$

Combining the terms, and defining $\bar{\mathbf{P}} \triangleq \mathbf{P}\mathbf{P}^\top$ and $\bar{\mathbf{A}} \triangleq \mathbf{A}\mathbf{A}^\top$ for convenience, the gradient of the reconstruction loss w/r/t the parts factors is thus

$$\nabla_\mathbf{P} \mathcal{L} = \sum_{i=1}^{N} -4\left(\mathbf{Z}_i^\top \mathbf{A}\mathbf{A}^\top \mathbf{Z}_i \mathbf{P}\right) + 2\left(\mathbf{P}\mathbf{P}^\top \mathbf{Z}_i^\top \mathbf{A}\mathbf{A}^\top \mathbf{A}\mathbf{A}^\top \mathbf{Z}_i \mathbf{P} + \mathbf{Z}_i^\top \mathbf{A}\mathbf{A}^\top \mathbf{A}\mathbf{A}^\top \mathbf{Z}_i \mathbf{P}\mathbf{P}^\top \mathbf{P}\right) \tag{10}$$

$$= 2\left(\sum_{i=1}^{N} \bar{\mathbf{P}}\mathbf{Z}_i^\top \bar{\mathbf{A}}\bar{\mathbf{A}}\mathbf{Z}_i \mathbf{P} + \mathbf{Z}_i^\top \bar{\mathbf{A}}\bar{\mathbf{A}}\mathbf{Z}_i \bar{\mathbf{P}}\mathbf{P} - 2\mathbf{Z}_i^\top \bar{\mathbf{A}}\mathbf{Z}_i \mathbf{P}\right). \tag{11}$$

Via similar arguments, the gradient w/r/t the channel factors is

$$\nabla_\mathbf{A} \mathcal{L} = \sum_{i=1}^{N} -4\left(\mathbf{Z}_i \mathbf{P}\mathbf{P}^\top \mathbf{Z}_i^\top \mathbf{A}\right) + 2\left(\mathbf{A}\mathbf{A}^\top \mathbf{Z}_i \mathbf{P}\mathbf{P}^\top \mathbf{P}\mathbf{P}^\top \mathbf{Z}_i^\top \mathbf{A} + \mathbf{Z}_i \mathbf{P}\mathbf{P}^\top \mathbf{P}\mathbf{P}^\top \mathbf{Z}_i^\top \mathbf{A}\mathbf{A}^\top \mathbf{A}\right) \tag{12}$$

$$= 2\left(\sum_{i=1}^{N} \bar{\mathbf{A}}\mathbf{Z}_i \bar{\mathbf{P}}\bar{\mathbf{P}}\mathbf{Z}_i^\top \mathbf{A} + \mathbf{Z}_i \bar{\mathbf{P}}\bar{\mathbf{P}}\mathbf{Z}_i^\top \bar{\mathbf{A}}\mathbf{A} - 2\mathbf{Z}_i \bar{\mathbf{P}}\mathbf{Z}_i^\top \mathbf{A}\right). \tag{13}$$

## A.1  ORTHOGONALITY AND CLOSED-FORM SOLUTION

Strictly speaking, Section 4.1 requires $\mathbf{A}$ to be an orthogonal matrix for the necessary equivalence $\mathbf{A}^\top \mathbf{Z} = \mathbf{A}^{-1}\mathbf{Z}$ to hold for the interpretation as a change of basis. In practice, we find our appearance factor matrix $\mathbf{A} \in \mathbb{R}^{C \times C}$ to be very close to orthogonal. For example, for BigGAN on the 'alp' class at layer 7, we find the mean element of $|\mathbf{A}^\top \mathbf{A} - \mathbf{I}_C|$ to be 8.33e−4. We show in Fig. 7 the 1$^{\text{st}}$ appearance vector localized in the image (following the procedure in Section 4.1.2) through both $\mathbf{A}^\top \mathbf{Z}$ and $\mathbf{A}^{-1}\mathbf{Z}$, where we see that, for the purposes of understanding the appearance factor visually as controlling the 'sky' concept, the two results are near-identical. Thus, we can take $\mathbf{A}$ to be an orthogonal matrix for our practical purposes.

If a more strict form of orthogonality is desired, one has the option to instead compute the appearance factors' solution in closed-form following Xu et al. (2005). In particular, let $\mathcal{Z} \in \mathbb{R}^{M \times C \times S}$ be the batch of mode-3-unfolded feature maps as in the main paper. Using the mode-$n$ product (Kolda & Bader, 2009), the partial multilinear projection of the feature maps onto the parts subspace is given by $\mathcal{Y} = \mathcal{Z} \times_3 \mathbf{P}^\top \in \mathbb{R}^{M \times C \times R_S}$. The appearance factors' solution is then given in closed-form by the leading eigenvectors of $\mathbf{Y}_{(2)}\mathbf{Y}_{(2)}^\top \in \mathbb{R}^{C \times C}$ (Xu et al., 2005). When using this closed-form solution, we find the mean element of $|\mathbf{A}^\top \mathbf{A} - \mathbf{I}_C|$ is 9.11e−5 in the same setting as above.

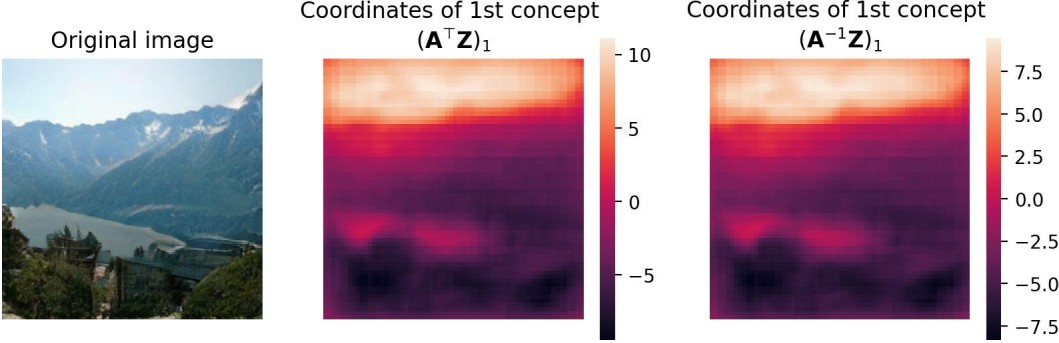

Figure 7: Comparing the concept localization with $\mathbf{A}^{\top}$ and $\mathbf{A}^{-1}$, we see that they are near-identical.

# B    ABLATION STUDIES

In this section, we present a thorough study of the various parts of our method, and the resulting learnt parts factors.

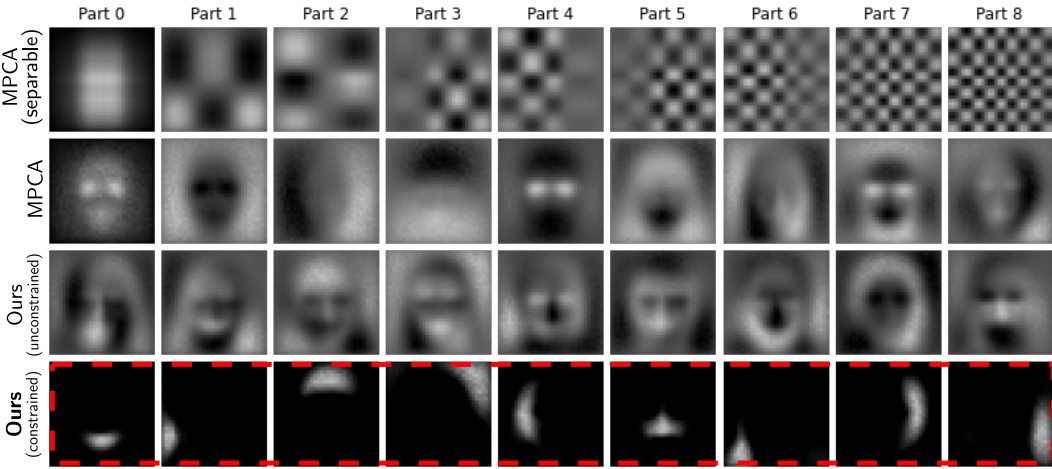

Figure 8: Ablation study comparing the parts factors learnt with various constraints and formulations. As can be seen, only our constrained formulation learns factors that span local parts-based semantic regions.

### B.0.1    CONSTRAINTS AND FORM

We first study the impact of the non-negativity constraints on the parts factors, and the importance of operating on the mode-3 unfolded $\mathbf{Z}_{i(3)} \in \mathbb{R}^{C \times H \cdot W}$ tensors (rather than their original $3^{\text{rd}}$-order form $\mathcal{Z}_i \in \mathbb{R}^{H \times W \times C}$). We show along the rows of Fig. 8 the resulting parts factors using various forms of decomposition and constraints. In particular, naively applying MPCA (Lu et al., 2008) (row 1) to decompose $\mathcal{Z}_i$ imposes a separable structure *between* the spatial modes, restricting its ability to capture semantic spatial regions. Moreover, even when combining the spatial modes and decomposing $\mathbf{Z}_{i(3)}$, the solution given by MPCA (Lu et al., 2008) (row 2) and by optimizing our method *without* any non-negativity constraints (row 3) leads to parts factors spanning the entire spatial window. This is due to the non-additive nature of the parts. However, as shown in row 4 of Fig. 8, only our constrained method successfully finds local, non-overlapping semantic regions of interest.

### B.0.2    PARTS FACTORS REFINEMENT

Finally, we showcase the benefit of our optional parts factors refinement process for data with no alignment. In row 2 of Fig. 9, we show the global parts factors overlaid over the target samples.

Clearly, for extreme poses (or in the case of data with no alignment, such as with animals and cars), these global parts will not correspond perfectly to the specific sample's parts (row 2 of Fig. 9). However, after a few projected gradient descent steps of Eq. (6), we see (row 3 of Fig. 9) that the refined parts factors span the specific parts of the individual samples more successfully. This refinement step is very fast; for example, at $l = 6$ it takes only 127ms (100 iterations).

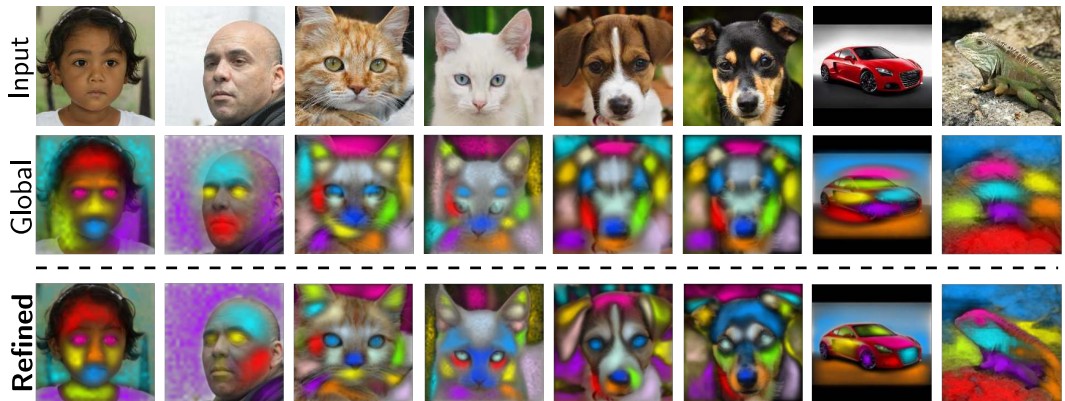

Figure 9: Visualization of the global parts factors (middle row) and the refined factors (bottom row) for particular samples (top row).

## C  ADDITIONAL EXPERIMENTAL RESULTS

In this section, we provide additional experimental results to support the main paper. We begin by presenting additional qualitative results on local image editing in Appendix C.1 and Appendix C.3. After this, we present additional ablation studies in Appendix C.6, and Appendices C.7.1 and C.7.2. Finally, we showcase additional experiments analyzing the runtime in Appendix C.8. However, we first provide a more detailed overview figure for our method–in Fig. 10 we show in more detail our proposed semi-nonnegative factorization in the context of a pre-trained image generator.

### C.1  TEST-TIME MANUAL PART SELECTION

In the main paper, we describe how *pixel-level* control is facilitated by our method in a way that is not possible with the SOTA (Zhu et al., 2021a; 2022). To demonstrate this, we show in Fig. 11 the resulting image when we add an appearance vector controlling the eye size to our spatial part for the eye region manually edited to cover over only *one* of the eyes. As we see, this clearly enlarges just a single eye, leaving the other untouched.

### C.2  BACKGROUND REMOVAL

Whilst not the main focus of the paper, we show that one can use the appearance vectors to identify the background of an image as one way of further demonstrating the quality of the concepts learnt and their utility for additional tasks. Prior work (Voynov & Babenko, 2020; Melas-Kyriazi et al., 2022) identify a "background removal" interpretable direction in the latent space of BigGAN (and GANs more generally in the case of Melas-Kyriazi et al. (2022)). By using our thresholded saliency maps for the "background" concept as a mask at the pixel-level, we can straightforwardly perform background *removal*.

For BigGAN, we show in Fig. 14a a comparison to Voynov et al. (2020) on their custom generator weights. For StyleGAN2, we make both a qualitative (Fig. 14b) and quantitative (Fig. 14c) to the mask predictions generated by the recent SOTA Melas-Kyriazi et al. (2022) on 4 different datasets. As can be seen, our method performs extremely well under the IoU metric, despite not being trained for this task. We generate "ground-truth" background/foreground masks using the general-purpose $U^2$ model Qin et al. (2020), which whilst not perfect, is often remarkably accurate as can be seen from the qualitative results.

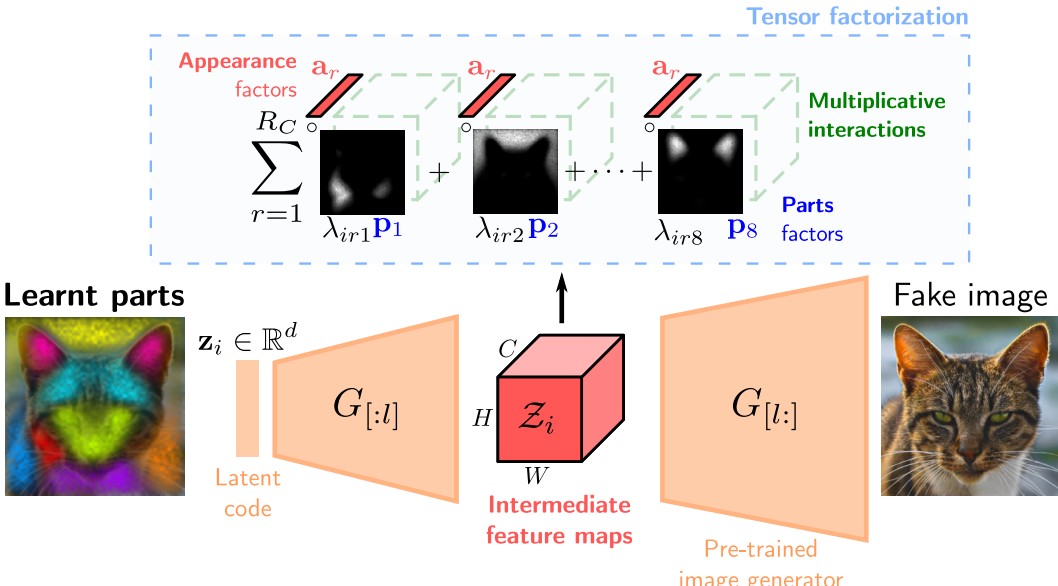

Figure 10: A more detailed overview of our method, in the context of the image generator. We apply our semi-nonnegative tensor decomposition on the intermediate convolutional feature maps, to extract factors for parts and appearances.

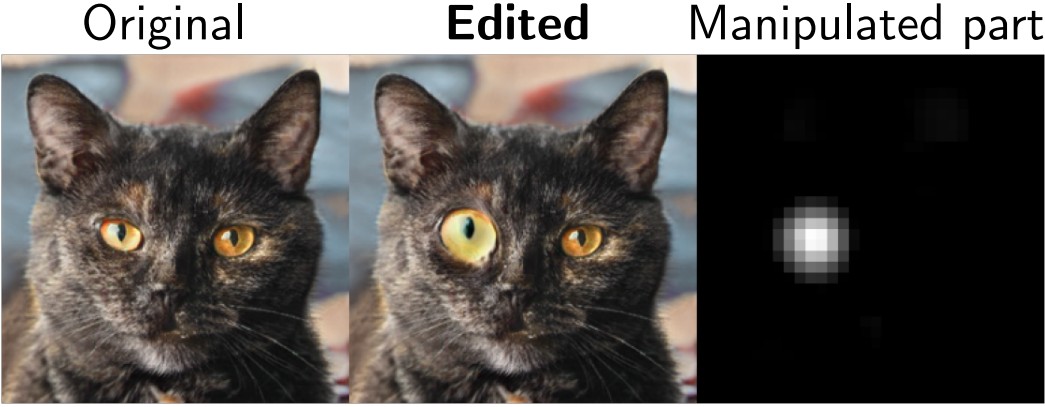

Figure 11: One can manually edit the learnt parts factors or provide custom ones for pixel-level control. For example, using half of the "eyes" part to affect only a single eye.

## C.3 ADDITIONAL QUALITATIVE RESULTS

Here we show many more qualitative results, on all 5 generators. Results for StyleGAN2 are shown in Fig. 12, and the other models in Fig. 13. As can be seen, our method is applicable to a large number of generators and diverse datasets. Additionally, we show in Fig. 15 that our method learns a diverse range of appearance vectors. Thus, it is possible to chain together a sequence of local edits, removing some objects, and modifying the appearance of other regions. This results in a powerful technique with which one can make complex changes to a scene of interest.

## C.4 ADDITIONAL COMPARISONS

We next include additional comparisons in Fig. 16 to the SOTA methods for all 4 local edits found in all baseline methods. As can be seen, whilst the SOTA local image editing methods excel at making prominent, photo-realistic changes to the ROI, they once again affect large changes to the image beyond just the ROI, as visualized by the MSE in the bottom rows. This is quantified once again with

Original    Part(s)    **Modified appearance**    Original    Part(s)    **Modified appearance**

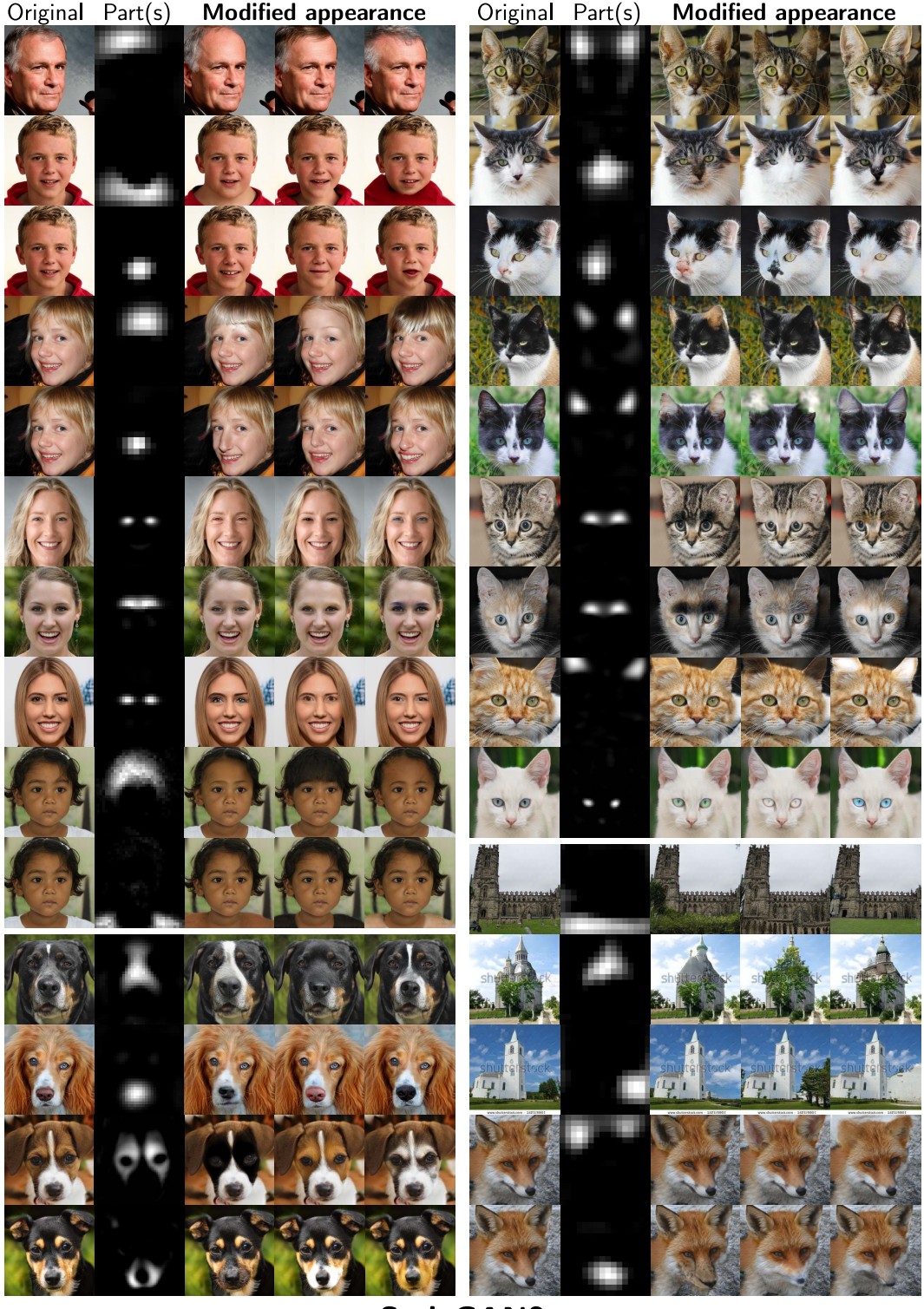

**StyleGAN2**

Figure 12: Additional local image editing results for StyleGAN2: at each column, we edit the target part with a different appearance vector.

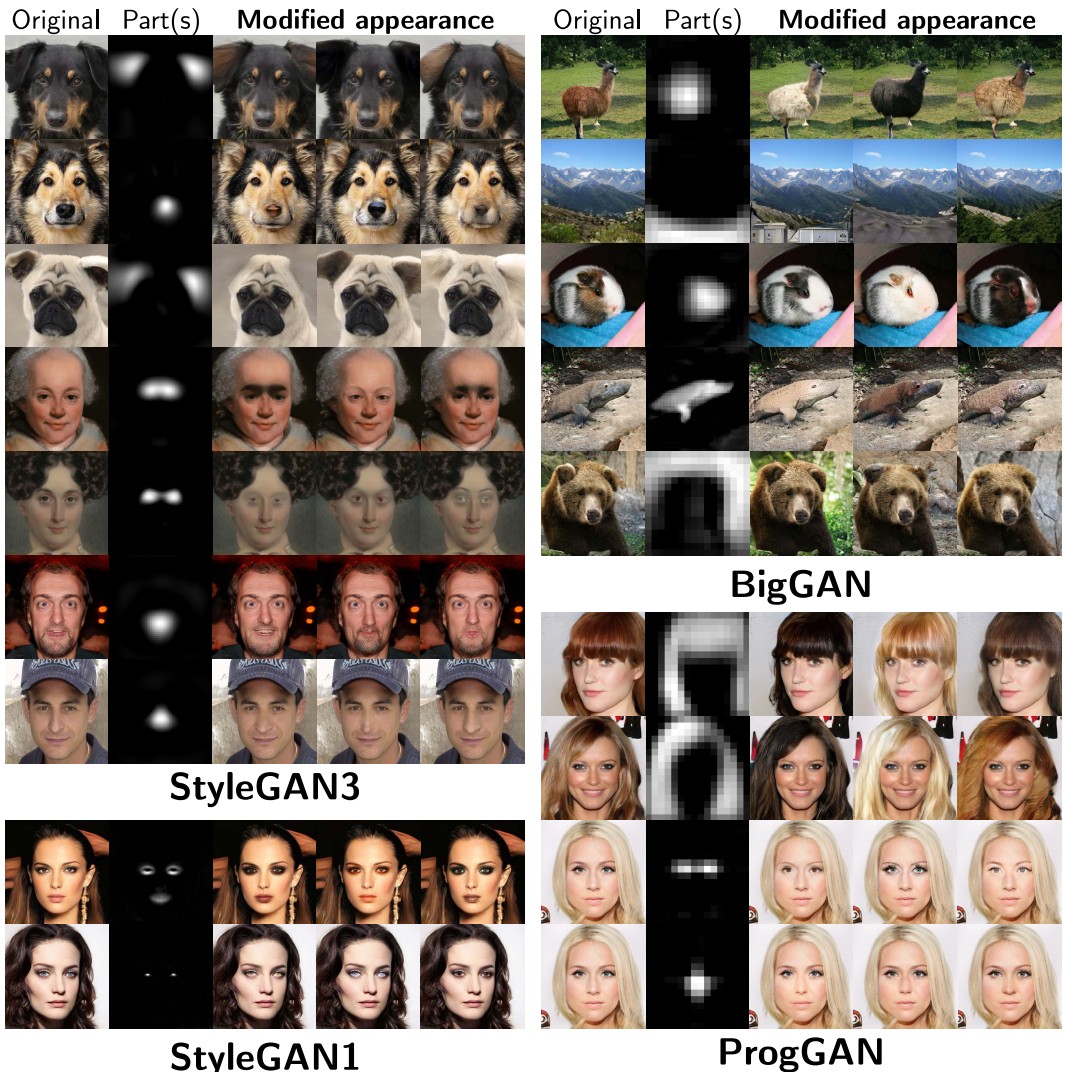

Figure 13: Additional local image editing results for 4 generators: at each column, we edit the target part with a different appearance vector.

the ROIR metric for six more local edits found in both our method and StyleSpace (Wu et al., 2021) in Table 3. We also show these images' quality is comparable to the SOTA by computing the FID (Heusel et al., 2017) metric in Table 4, for 10k samples per edit.

Table 3: Additional ROIR (↓) results for six more local edits (1k images per edit)

|  | Wide nose | Dark eyebrows | Light eyebrows | Glance left | Glance right | Short eyes |
|---|---|---|---|---|---|---|
| StyleSpace | $2.52 \pm 0.91$ | $1.99 \pm 1.28$ | $1.95 \pm 1.13$ | $1.54 \pm 1.14$ | $1.50 \pm 1.17$ | $1.91 \pm 1.41$ |
| **Ours** | $\mathbf{0.80} \pm 0.23$ | $\mathbf{0.51} \pm 0.19$ | $\mathbf{0.41} \pm 0.13$ | $\mathbf{0.68} \pm 0.24$ | $\mathbf{0.67} \pm 0.24$ | $\mathbf{0.46} \pm 0.13$ |

## C.5 SALIENCY MAPS

In this section we show additional results on saliency map generation. In Fig. 18 we show this for the "background" concept, which can be used to remove the background in the image, with the negative mask. We also provide localization results on additional concepts such as "sky" and "concrete" in

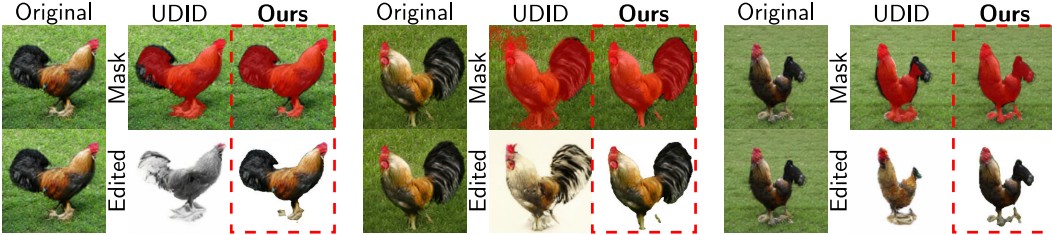

(a) A comparison of our method to Voynov & Babenko (2020) for background removal and object detection: we leave the object of interest largely untouched.

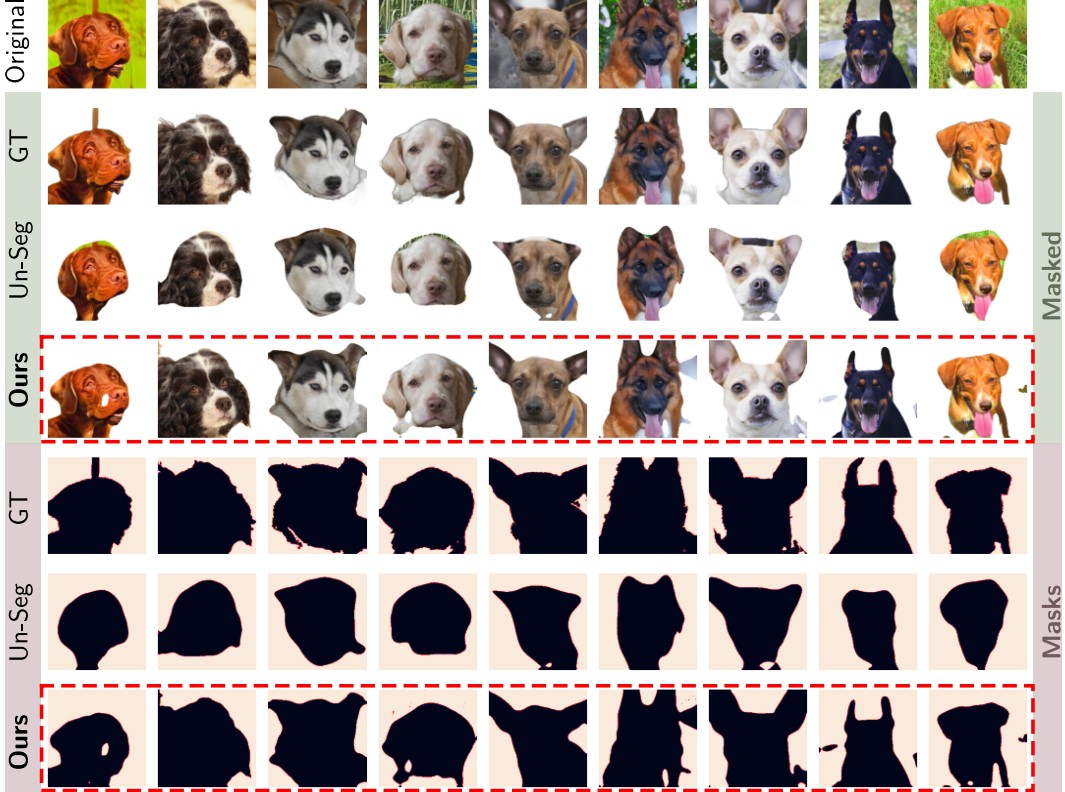

(b) Qualitative comparison on StyleGAN2 to Melas-Kyriazi et al. (2022) for AFHQ-dogs. The single dataset-specific background appearance factor can often accurately segment the images.

| | AFHQ-dogs | | AFHQ-cats | | FFHQ | | MetFaces | |
|---|---|---|---|---|---|---|---|---|
| | Mean IoU | Median IoU | Mean IoU | Median IoU | Mean IoU | Median IoU | Mean IoU | Median IoU |
| Melas-Kyriazi et al. (2022) | $0.48 \pm 0.22$ | 0.52 | $0.34 \pm 0.18$ | 0.30 | $0.37 \pm 0.23$ | 0.36 | $0.28 \pm 0.27$ | 0.21 |
| Ours | $\mathbf{0.65 \pm 0.19}$ | **0.72** | $\mathbf{0.54 \pm 0.19}$ | **0.58** | $\mathbf{0.46 \pm 0.19}$ | **0.47** | $\mathbf{0.52 \pm 0.19}$ | **0.53** |

(c) Quantitative comparison to Melas-Kyriazi et al. (2022) for StyleGAN2 on 4 datasets, using the author's official code, and recommended $r = 0.2$. We train for the default 300 iterations on all datasets apart from the two face-based datasets, where we find their method to give much better results at 100 iterations. For each dataset we compute the IoU over 1k samples.

Figure 14: Demonstrating the appearance factor's semantics through background mask generation.

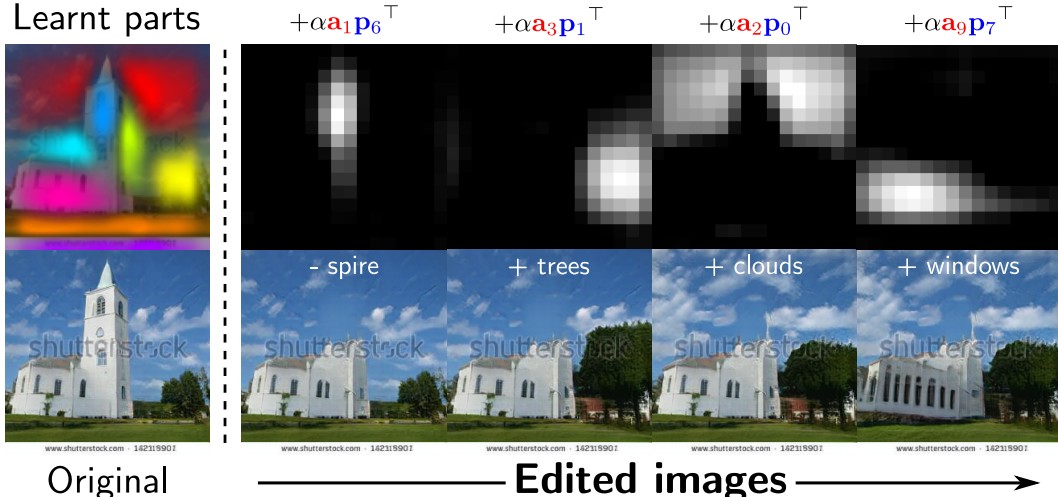

Figure 15: Progressively editing the feature maps with appearance vectors at various regions. Ultimately one can perform complex composite changes to a scene.

Table 4: FID ($\downarrow$) (Heusel et al., 2017) on 10k FFHQ samples per local edit.

|  | Eyes | Nose | Open mouth | Smile |
|---|---|---|---|---|
| GANSpace (Härkönen et al., 2020) | 29.92 | 30.60 | 46.85 | 30.62 |
| SeFa (Shen & Zhou, 2021) | 30.23 | 31.52 | 31.73 | 29.53 |
| StyleSpace (Wu et al., 2021) | 29.81 | 29.62 | 31.47 | 30.19 |
| LowRankGAN (Zhu et al., 2021a) | 30.57 | **29.18** | 30.95 | 30.60 |
| ReSeFa (Zhu et al., 2022) | **29.37** | 30.02 | 29.63 | 29.45 |
| **Ours** | 29.91 | 29.77 | **29.26** | **28.91** |

Fig. 20. Interestingly, our method learns concepts at 3 layers of depth in BigGAN. For example, we see foreground, mid-ground, and background concepts emerge independently as visualized in Fig. 19.

## C.6 DECOMPOSITION RANK

We next demonstrate the impact of the choice of rank $R_S$ for the parts factors. Shown in Fig. 21 are various *global* parts overlaid over a random image from the dataset. As can be seen, a smaller value of $R_S$ leads to larger semantic parts such those spanning the entire face, whilst a larger value of $R_S$ leads to spatially smaller parts such as the eyebrows and teeth. This can in essence be viewed as a hyperparameter that affords a user control over the size of the parts one wishes to learn. Additional comparisons are shown for many datasets in Figs. 22 to 24.

## C.7 INITIALIZATION STRATEGIES

### C.7.1 PARTS FACTORS INITIALIZATION

We find our method is robust to different non-negative initialisations of the parts factors. This is a crucial benefit of our method–we do not require an SVD-based initialisation (Cichocki et al., 2009; Boutsidis & Gallopoulos, 2008; Yuan et al., 2009) for the parts factors, and thus need not solve the corresponding $S$-dimensional eigenproblem in this step. As a case in point, at layer 10 in an 18-layer StyleGAN (Karras et al., 2019), an 16384-dimensional eigenproblem must be solved solely for the parts factors initialization–thus going deeper quickly becomes infeasible without the flexibility of the proposed method. Instead, our formulation permits initialization through sampling each element of the parts factors $\mathbf{P}$ from a random uniform distribution on the interval $[0, 0.01]$. This allows us to perform our decomposition at much later layers in the network than would otherwise be possible, and

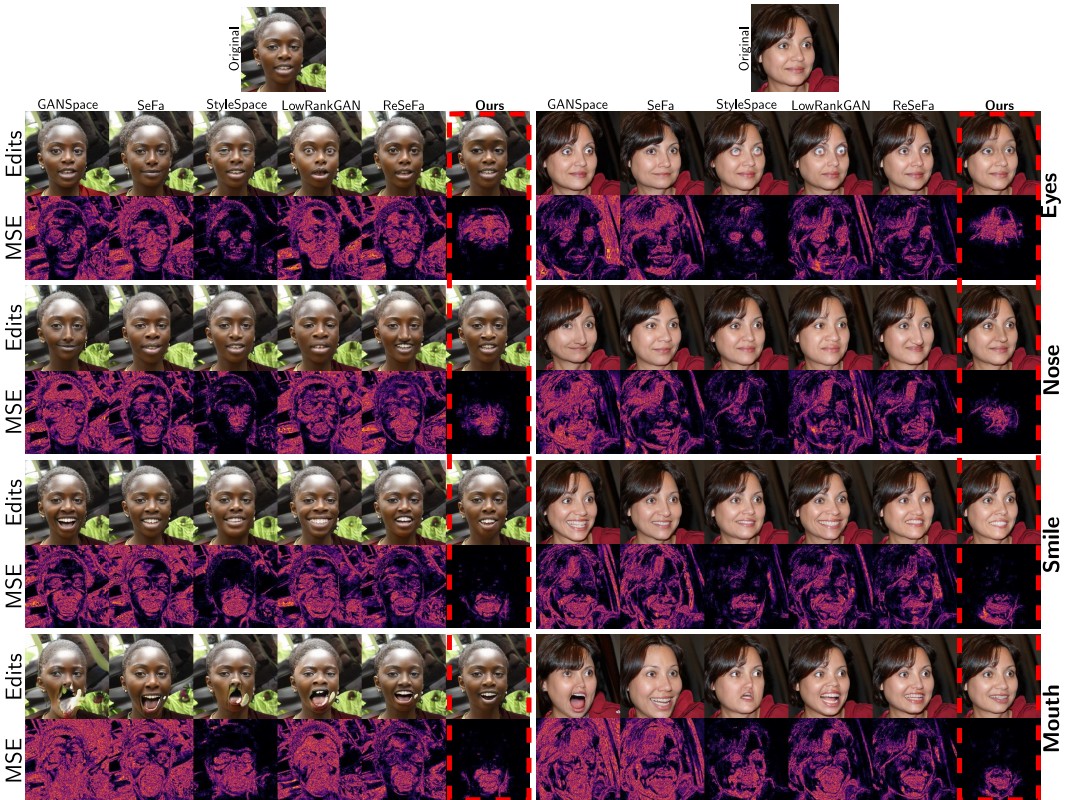

Figure 16: Qualitative comparison to SOTA methods for 4 local edits. As can be seen, our method succeeds in keeping the area outside the ROI unchanged, as intended.

consequently one can discover more fine-grained appearance vectors (due to the way in which later layers in StyleGAN control more fine-grained styles (Karras et al., 2018)).

### C.7.2 CHANNEL FACTOR INITIALIZATION

We find the leading left-singular vectors of the channel mode's scatter matrix to contain particularly semantically meaningful directions. This initialization is what allows us to locate common concepts such as "background" with ease, due to the ordering of the singular vectors by their singular values.

Intuitively, the leading few left-singular vectors for the channel mode's scatter matrix capture frequently occurring appearances and textures. For faces, this is largely textures such as skin, and hair. Whilst for other datasets, common textures include the sky or floor. We show additional examples of this in Fig. 25, where the first columns of the appearance basis (through our initialization) correspond to the "background" and "skin" texture. For example, one can remove the facial features of a person by adding this appearance vector, seen in Fig. 25 (b). Whilst we find our decomposition also works with random initialization of the channel factors, the first few vectors do not necessarily correspond to the more frequently appearing concepts in the same way the SVD-based initialization provides, meaning they are less easy to interpret.

### C.8 RUNTIME AND OBJECTIVE

An important benefit of our method is the lack of need to compute expensive gradient maps or Jacobians with respect to target regions, as is required in Zhu et al. (2021a); Wu et al. (2021); Zhu et al. (2022). To quantify this, we plot in Fig. 26 the total training time required to train the SOTA methods to produce the four local directions used in the main paper. In particular, we train the methods for all 3 regions of interest ("mouth", "eyes", and "nose"), with a single Quadro RTX 6000

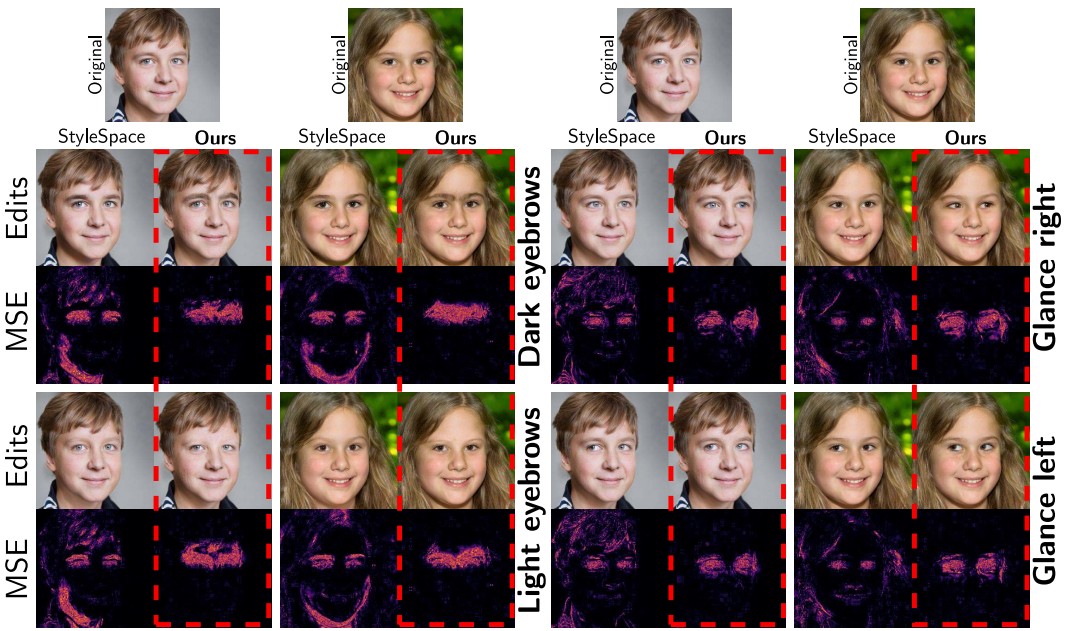

Figure 17: Qualitative comparison to more local edits for the StyleSpace method.

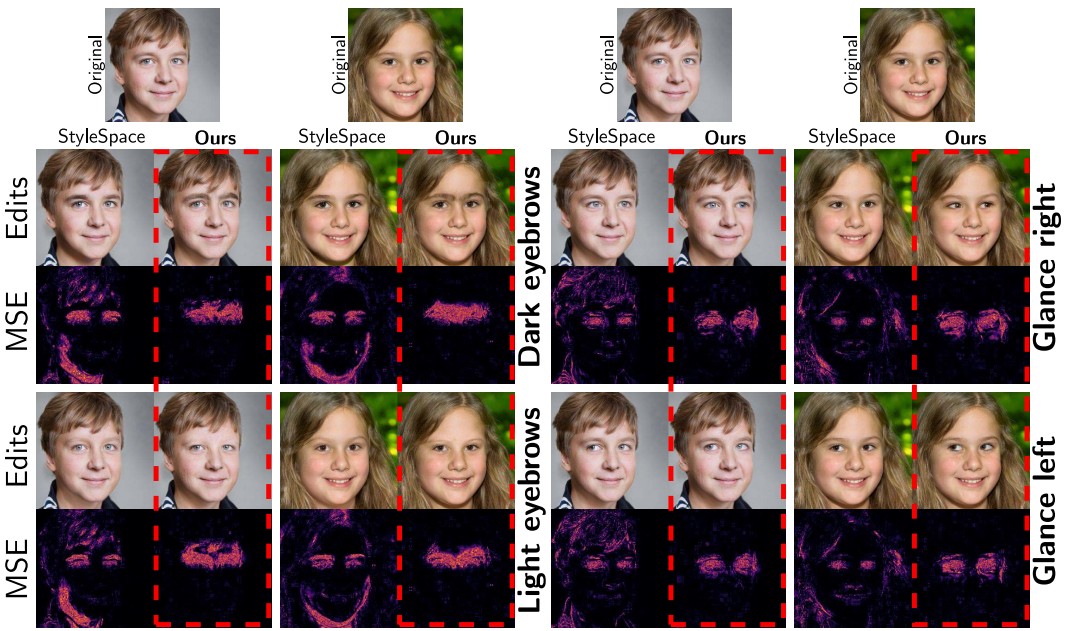

Figure 18: Saliency maps, generated masks, and subsequently removed backgrounds on AFHQ Dog, with StyleGAN2.

GPU, using the official authors' codebases[2]. We find that our method takes less than $1/400^{th}$ of the training time of LowRankGAN (Zhu et al., 2021a), and $1/170^{th}$ the time of StyleSpace (Wu et al., 2021)–greatly speeding up the task of local image editing.

Alternatively, one can use an optimizer such as Adam (Kingma & Ba, 2014) in an autograd framework (e.g., PyTorch (Paszke et al., 2019)) to compute Algorithm 1. We find this removes some sensitivity

---

[2]**StyleSpace**: https://github.com/betterze/StyleSpace,
**LowRankGAN**: https://github.com/zhujiapeng/LowRankGAN,
**ReSeFa**: https://github.com/zhujiapeng/resefa

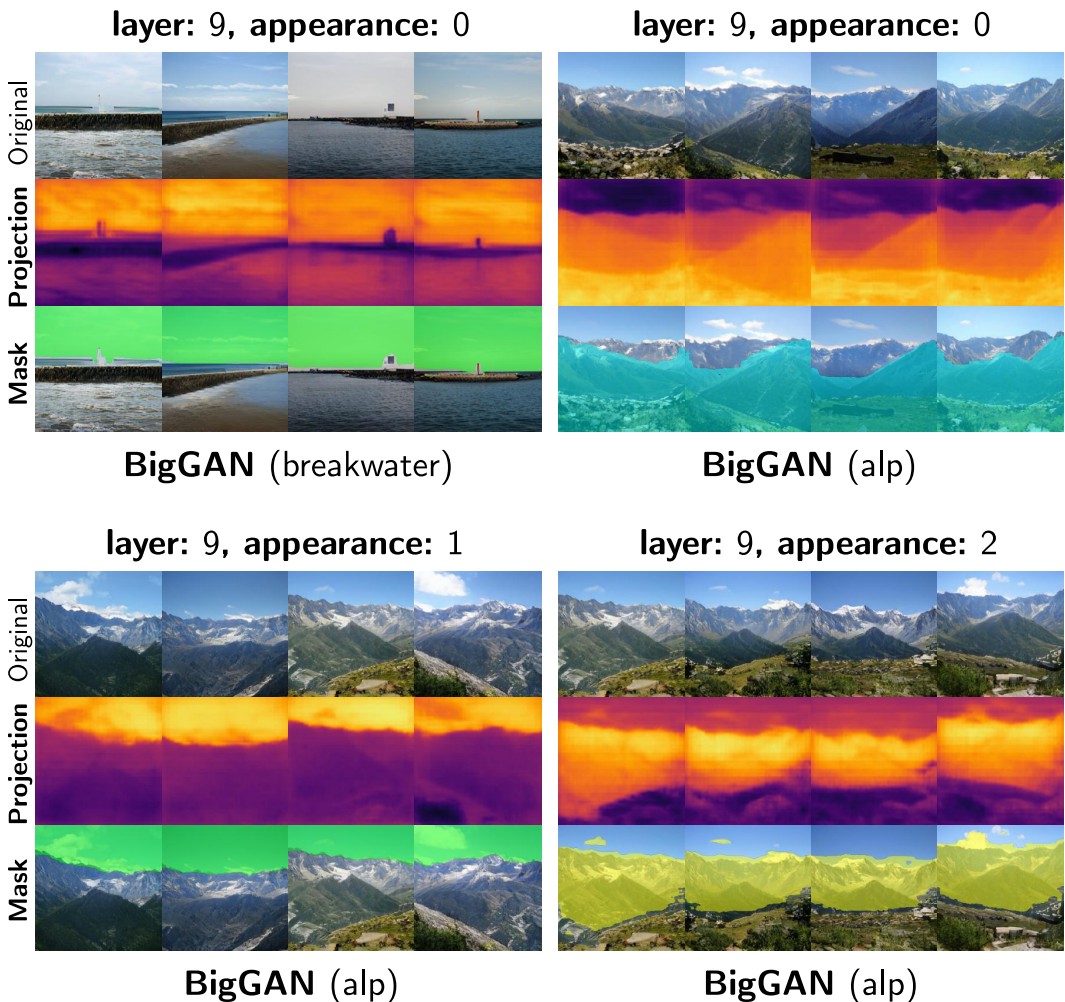

Figure 19: Saliency maps and generated masks on BigGAN "alp" and "breakwater". We see concepts at 3 different levels of depth emerge.

to the learning rate that comes with vanilla gradient descent. However, we find that when solving our objective manually with the gradients in Eq. (12), our method takes less than $1/3^{\text{rd}}$ of the time to train. This is a particularly useful performance boost when decomposing later layers in the network.

### C.8.1 RUNTIME ABLATIONS

To isolate the impact of the appearance factor training, in Fig. 27 we show the importance of descending the gradient in the channel subproblem (after 1000 initial steps of the parts subproblem) on top of the HOSVD initialisation, for 1000 samples. As can be seen, the cost strictly decreases as a function of the number of iterations $T$, far below that of the HOSVD initialisation.

We also show in Fig. 28 the impact of the number of training iterations $T$ for various choices of rank $R_S$. As can be seen, the model is relatively stable over iterations in assigning global parts factors.

### C.9 IMPLEMENTATION DETAILS

We use a modified version of the GenForce (Shen et al., 2020b) library for the ProgressiveGAN, StyleGAN1, and StyleGAN2 models[3]. We also use the TensorLy (Kossaifi et al., 2019) library for

---

[3]in addition to **StyleGAN3**: https://github.com/NVlabs/stylegan3,
**BigGAN**: https://github.com/huggingface/pytorch-pretrained-BigGAN.

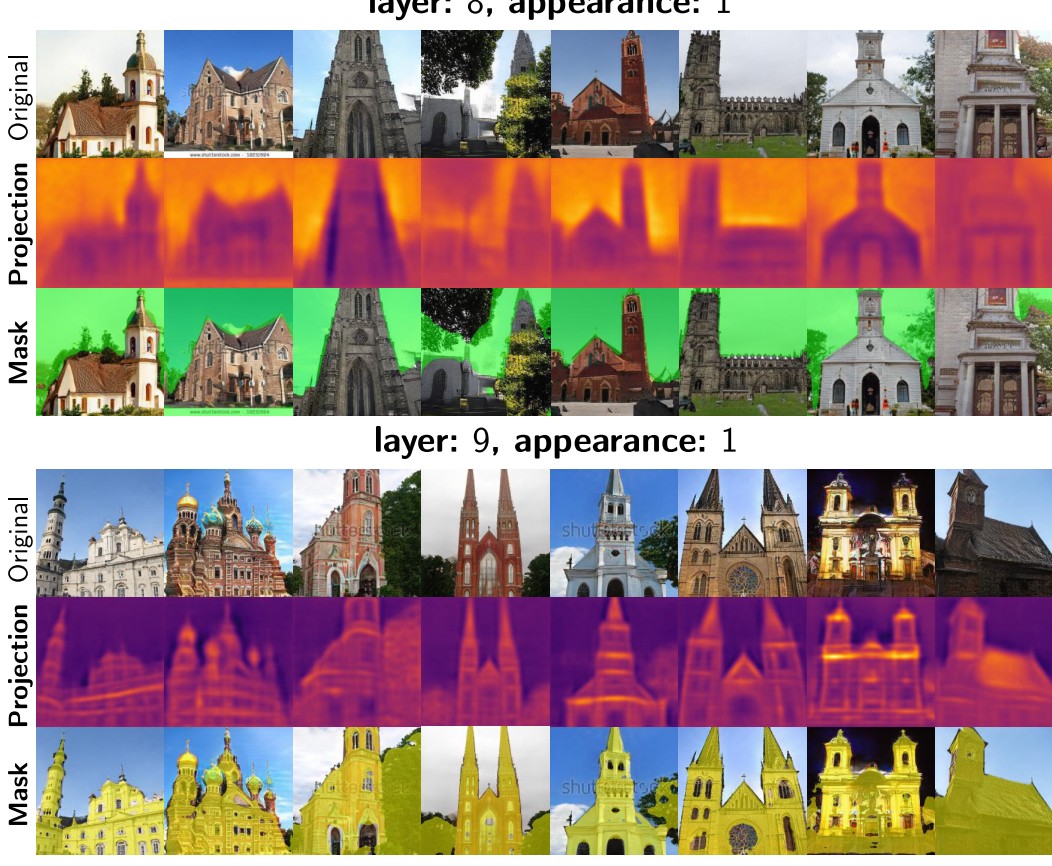

Figure 20: Saliency maps and generated masks on StyleGAN2 (church). Shown here are what one could deduce to be 'sky' and 'concrete' concepts.

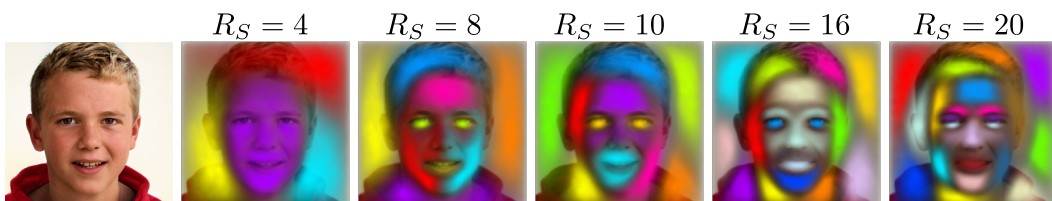

Figure 21: Learnt global parts factors for varying ranks $R_S$, with StyleGAN2 trained on FFHQ at layer $l = 9$, for 2000 iterations. As can be seen, *shared* semantic regions such as the eyes, the nose, and the mouth are captured by the parts factors.

the autograd implementation of our method. In practice, we find it useful (in terms of run-time) to descend the gradients of our loss function stochastically, sampling new activations each iteration.

### C.9.1  BASELINES

For both the quantitative and qualitative results for the baseline methods, we use the following directions annotated from the pre-trained models by the authors, where available:

- **GANSpace** (Härkönen et al., 2020): we use the following author-annotated directions: `Eye_Openness`, `Nose_length`, `Screaming`, and `Smile`.

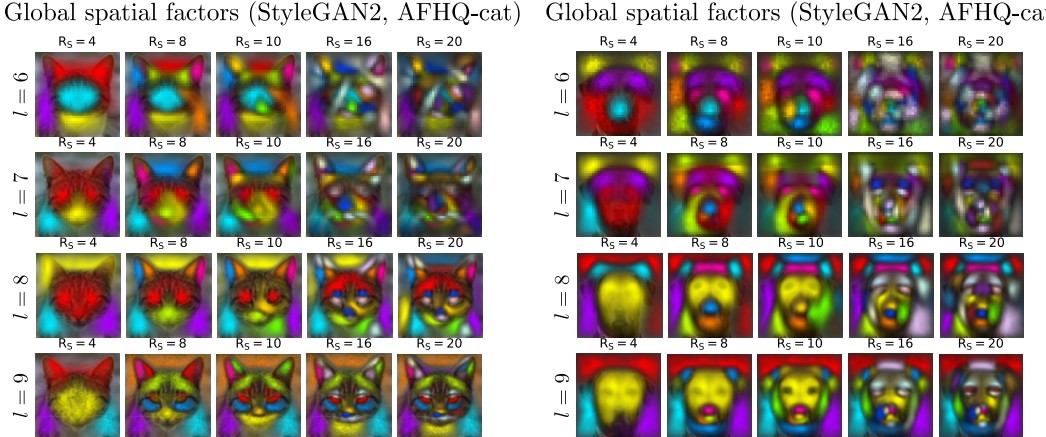

Figure 22: Learnt global parts factors for varying ranks $R_S$, with StyleGAN2 trained on two separate splits of AFHQ at various layers $l$. (zoom for detail). As can be seen, common semantic regions such as the eyes, the nose, and the mouth are captured by the parts factors.

# Global spatial factors (StyleGAN1, CelebA-HQ)

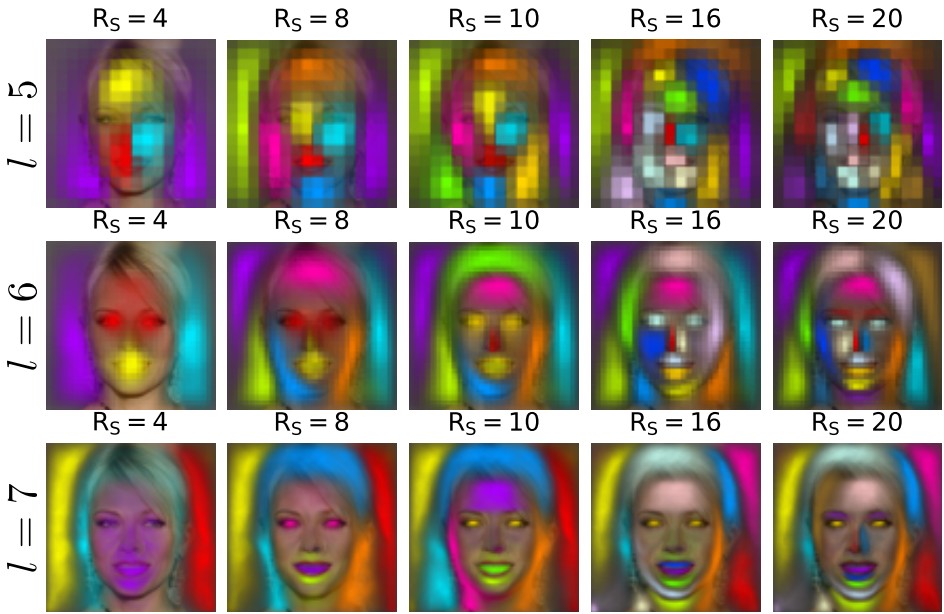

Figure 23: Learnt global parts factors for varying ranks $R_S$, with StyleGAN1 trained on CelebA-HQ at various layers $l$. As can be seen, common semantic regions such as the eyes, the nose, and the mouth are captured by the parts factors.

# Global spatial factors (StyleGAN2, FFHQ)

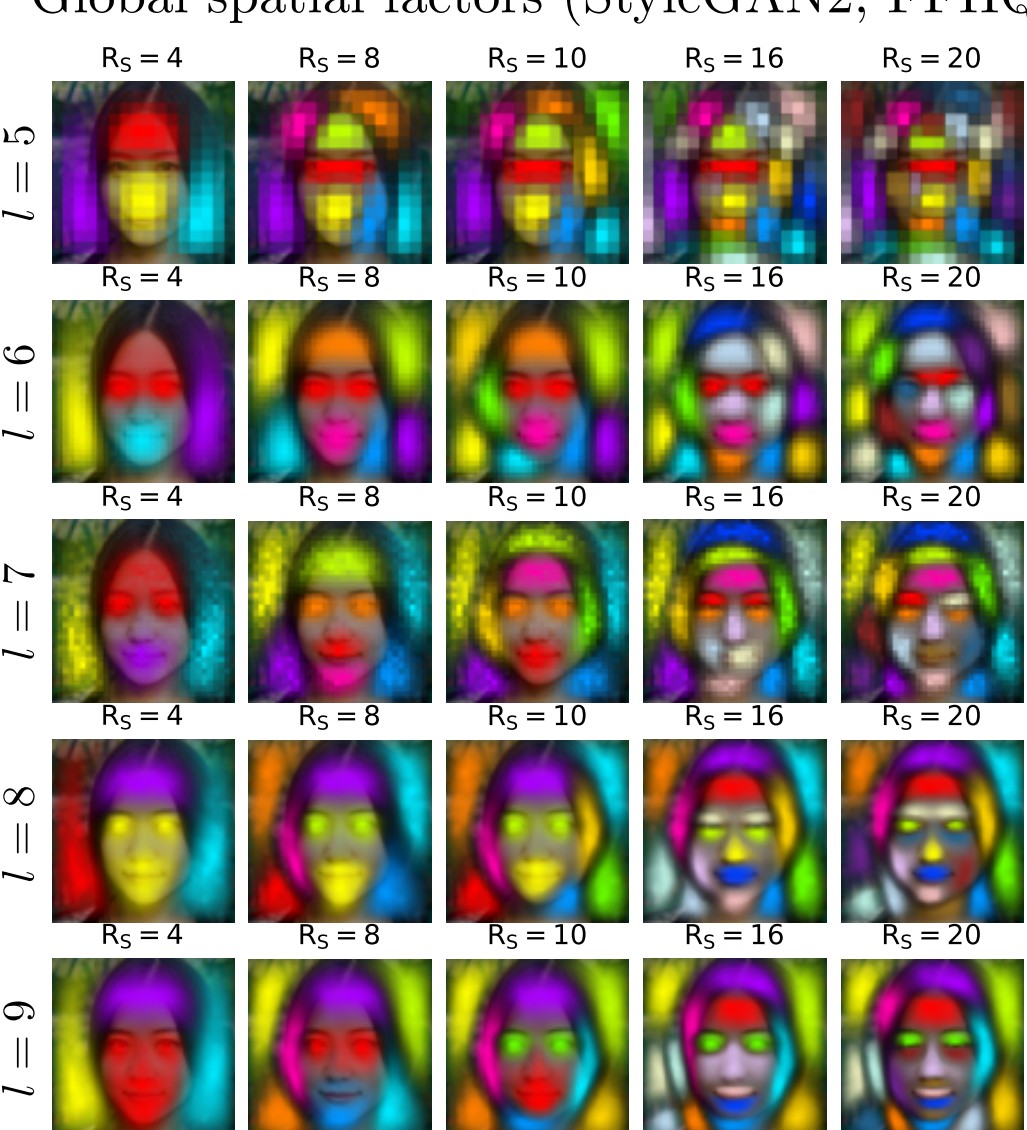

Figure 24: Learnt global parts factors for varying ranks $R_S$, with StyleGAN2 trained on FFHQ at various layers $l$. As can be seen, common semantic regions such as the eyes, the nose, and the mouth are captured by the parts factors.

- **SeFA** (Shen & Zhou, 2021): we use directions manually found by ourselves that most closely resemble the target attributes.

- **LowRankGAN** (Zhu et al., 2021a): we use the following author-annotated directions: `expression`, `eye_size`, `mouth_open`, `nose`.

- **StyleSpace** (Wu et al., 2021): we use following author-annotated directions: `8_289`, `6_113`, `6_202`, and `14_239`, where `x_y` is channel $x$ at generator level $y$ as described in (Wu et al., 2021).

- **ReSeFa** (Zhu et al., 2022): we use the following author-annotated directions: `eyesize`, `mouth`, and `nose_length`. We manually find a direction that most closely resembles a smile.

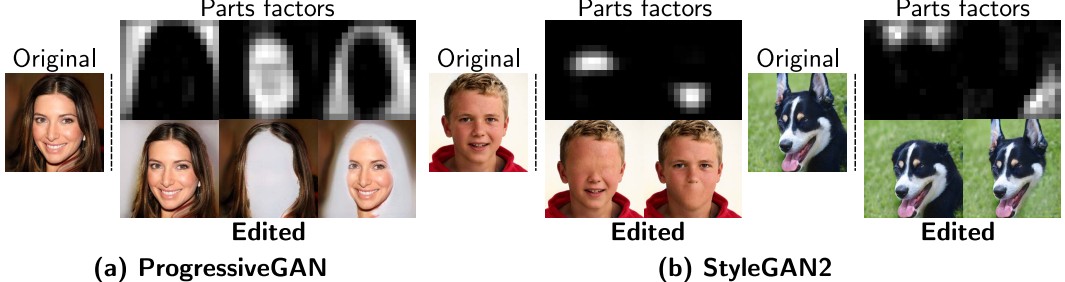

Figure 25: The appearance vectors obtained via an SVD-based initialization are particularly semantically meaningful: for example, the 1st column in the basis of PGGAN removes objects from the image in CelebA-HQ, whilst the 2nd column for StyleGAN2 adds skin in FFHQ.

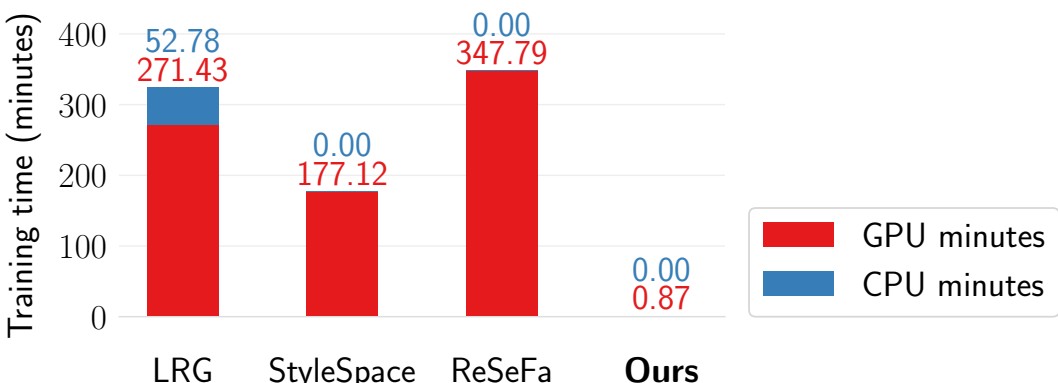

Figure 26: Total training time (to train the models used for the quantitative results) for our method and the SOTA, using a Quadro RTX 6000.

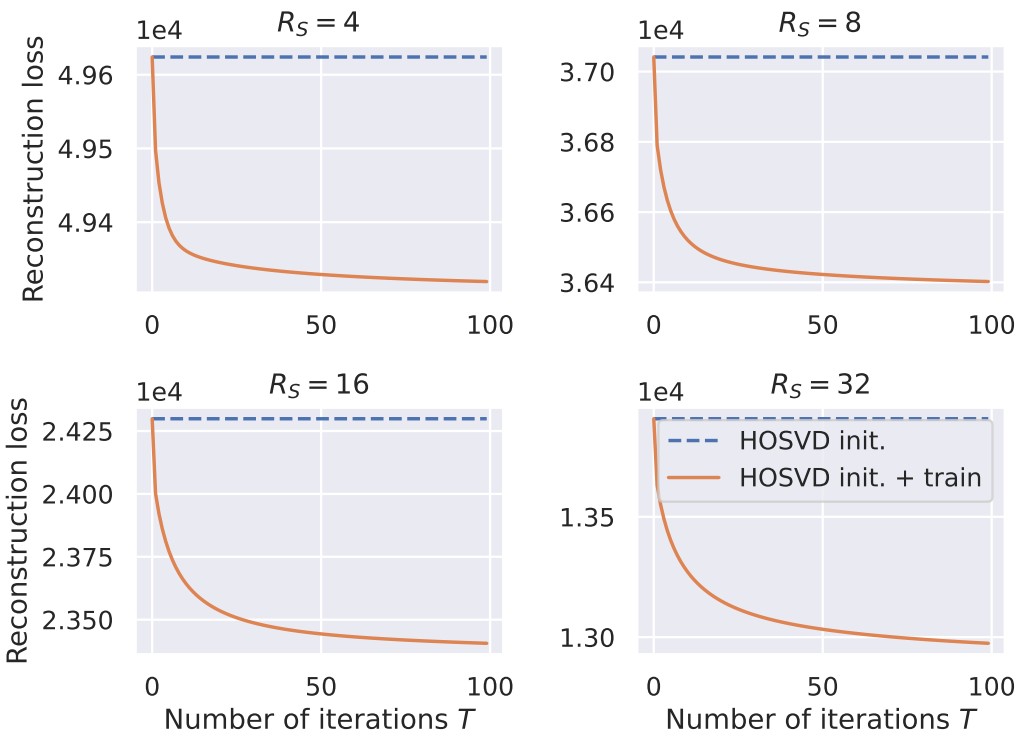

Figure 27: Appearance factor subproblem when using the fixed HOSVD initialization VS additionally descending the gradient, as a function of $T$ and for various number of parts $R_S$.

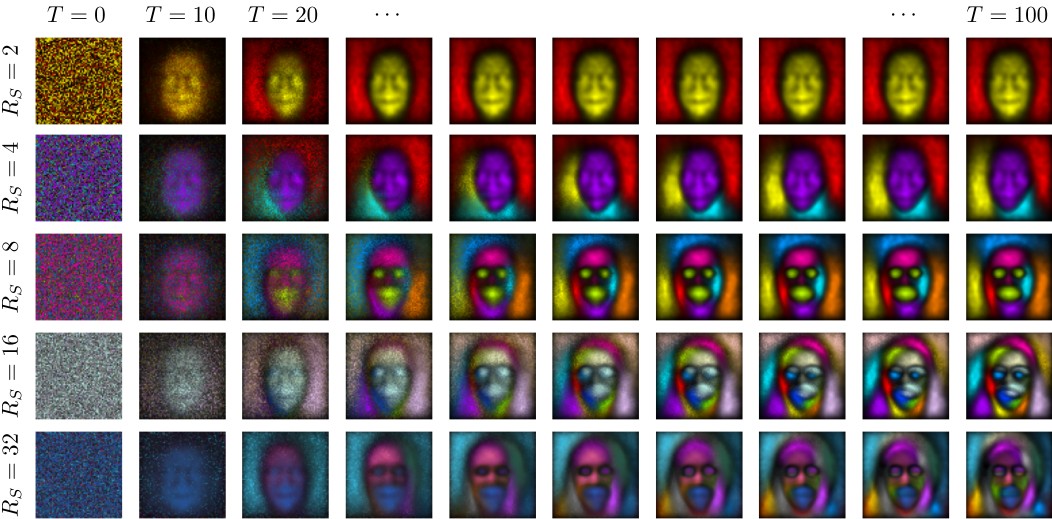

Figure 28: The learnt global parts factors as a function of training iterations $T$ for various choices of rank $R_S$: the parts factors are stable over iterations and reliably correspond to semantic parts across different values of $R_S$.

