# OpenReview forum: "PandA: Unsupervised Learning of Parts and Appearances in the Feature Maps of GANs"
_ICLR.cc/2023/Conference — ICLR 2023 poster_

### Official Review · Reviewer_zd35 · 2022-10-21

**Confidence:** 3
**Correctness:** 3
**Technical Novelty And Significance:** 2
**Empirical Novelty And Significance:** 3
**Recommendation:** 8

**Clarity, Quality, Novelty And Reproducibility:**

The paper is missing several many important explanations to understand the method. It is not explained how the editing works. Does the user inspect all discovered parts and then edit their spatial or semantic components? How does changing semantics work? Where does the changed appearance vector come from?

**Details Of Ethics Concerns:**

Ethics concerns have been added

**Strength And Weaknesses:**

## Strengths
The method is very fast and simple as it is based on a linear decomposition of feature maps of the generator. This means that no network training is needed to discover the parts and appearances.

It is possible to finetune the decomposition for individual images to allow for more precise edits. Additionally, as the decomposition is spatial, the edits can be more targeted that editing the latent space of the GAN alone.

The method has been evaluated on a range of GAN architectures and datasets which shows the generality of the method and its independence from specific generators.

## Weaknesses

A main weakness of the paper is the evaluation. Although evaluating semantic editing of GAN generated images is a difficult problem, it is necessary to support the claims of the paper.
### W1 Semanticity
It is not clear if the part decomposition is indeed semantic and not spatial. It is likely that the linear decomposition prefers decomposing into spatial regions and relies on the alignment of samples across the dataset. See e.g. Fig 9 columns 4,7,8 where instance alignment is not present. For conditional GANs, can semantic parts be derives across classes to find correspondences between parts (e.g. to replace dog legs with cat legs)?

### W2 Background
The paper emphasizes the ability of the model to identify the background/foreground of images. The fact that this is possible has been shown not only by (Voynov&Babenko 2020) but also in (Melas-Kyriazi et al, 2021) for a large variety of generative models. The quality of the background discovery could be measured possibly in two ways. Either through inversion of the GAN on real images with associated ground truth on a saliency segmentation dataset or relative to a supervised model, directly on the generated images. This would be helpful, as currently the results look more similar to a simple spatial decomposition rather than a semantic one. (e.g. c,d in Fig 4, Fig 19) similar to the decomposition on face datasets seemingly strongly relying on the alignment between images.

### W3 Hyper-parameter and ablation
The method contains several hyper-parameters and specific choices. It would be valuable to understand their influence on the model. E.g.: How well does the model work just with SVD alone? What is the influence of the number of iterations T? Are there substantial differences between different GANs trained on the same data - do certain architectures lead to feature maps that are easier to decompose? Can appearance vectors of the same part be swapped between instances? How does the layer choice matter? Does the user need to select the layer for decomposition or is there an automated way to choose? Other forms of clustering could be compared, e.g. k-means.


**Summary Of The Paper:**

The paper proposes a method to decompose the feature maps of a GAN generator into spatial and semantic components. This decomposition allows editing the output image at the level of parts. The experiments show that the method can be applied to several different GAN architectures and datasets. Additionally, due to the simplicity of the approach, the method is very fast.

**Summary Of The Review:**

Overall the paper presents a simple approach to enable more fine-grained editing of GAN images by decomposing feature maps. Despite the strengths, the current state of the evaluation makes it difficult to assess the benefits of this method over existing techniques.

---

> ### Author Response · Authors · 2022-11-17
> **Response to Reviewer-zd35 [1/N]**
>
> We thank Reviewer-zd35 for their detailed review. However, we respectfully disagree with the reviewer’s comment in the summary:
>
>
>   > "the current state of the evaluation makes it difficult to assess the benefits of this method over existing techniques."
>
> for the reason that “background removal” (the task with which Reviewer-zd35 is concerned) is one small additional experiment in the supplementary material only. We already present thorough comparisons in the main paper to our primary stated task of local image editing, including the recent SOTA from ICML'22 [1]. This is corroborated by the other reviewers’ responses:
>
>
> - **Reviewer-WFbU** lists our experimental comparisons as a key strength of the paper: `"localized editing is well demonstrated qualitatively, and also quantitative evaluated with an appropriate evaluation metric".`
> - **Reviewer-3WTy** states: `"the authors made a good job at proving their contribution experimentally and providing a detailed analysis of the method through ablation studies".`
>
> Nevertheless, we have conducted the requested experiments on 4 datasets for background segmentation in comparison to the requested work [2] from ICLR'22 on 4 datasets, where our method consistently performs the best under the IoU metric for StyleGAN2. Please see Fig. 14 in the supplementary material for more details and for both qualitative and quantitative results. The quantitative results are included here for reference:
>
> |                       | AFHQ-dogs        |            | AFHQ-cats        |            | FFHQ             |            | MetFaces         |            |
> | --------------------- | ---------------- | ---------- | ---------------- | ---------- | ---------------- | ---------- | ---------------- | ---------- |
> |                       | Mean IoU         | Median IoU | Mean IoU         | Median IoU | Mean IoU         | Median IoU | Mean IoU         | Median IoU |
> | Uns. Seg. [2] | 0.48 +- 0.22     | 0.52       | 0.34 +- 0.18     | 0.30       | 0.37 +- 0.23     | 0.36       | 0.28 +- 0.27     | 0.21       |
> | **Ours**              | **0.65 +- 0.19** | **0.72**   | **0.54 +- 0.19** | **0.58**   | **0.46 +- 0.19** | **0.47**   | **0.52 +- 0.19** | **0.53**   |
>
> We agree with the reviewer that such results further attest to the quality of our learnt factors, and we hope as **Reviewer-zd35** suggests, that this helps resolve the "semantic/spatial" concerns stated in W1. We hope that the reviewer will reconsider their recommendation based on this.
>
> **We further address all questions raised in all 3 points below:**
>
> 1. We recall that the parts factors are *semantic* in the sense that they learn to span spatial positions with semantically similar channel fibers (for example, see that our parts span both eyes, ears, or eyebrows in Fig. 1). We use “semantic similarity” here the same way it is used in the literature [3], i.e. to refer to channel fibers that lie over the same high-level concept in the image, such as the same facial feature of a face.
>
>     > can semantic parts be derived across classes to find correspondences between parts (e.g. to replace dog legs with cat legs)?
>
>     The proposed method can identify shared concepts across classes: e.g. in conditional BigGAN, "sky" and "foreground" factors are examples of some of the concepts found jointly across classes of outdoor scenes. One can achieve something like what the reviewer asks in our framework by “rewriting the concepts”: let $\mathbf{Z}_i = \mathbf{A}\Lambda_i$ for the $i$th sample’s feature maps. Then, $\Lambda_i(c, s)$ specifies how much of appearance vector $c$ is present at spatial position $s$ in the $i^\text{th}$ sample’s feature maps. With this understanding, one can swap/manipulate the rows of the coefficients $\Lambda_i\in\mathbb{R}^{C \times S}$ to change *which* appearance vectors are being placed at the desired spatial locations/parts (intuitively: e.g. at the spatial positions where it’s currently specified to write an appearance of a dog’s face, modify the coefficients to instead write there the appearance of a cat’s face).
>
> 2. Whilst we do use the "background" factor as an example of one useful concept learnt by our model, we kindly remind Reviewer-zd35 that the task of background segmentation is not at all the primary focus of the paper. Rather, our main focus is the **ways in which one can use these concepts for local image editing**--such as removing objects from an image or changing the style of a facial attribute. Nevertheless, as described above we have run the requested comparison, and we include the new experiments in the supplementary material in Fig. 14.
>
> **[continued below]**

---

> > ### Author Response · Authors · 2022-11-17
> > **Response to Reviewer-zd35 [2/2]**
> >
> > 3. We gently remind the reviewer that our submitted manuscript is already close to 30 pages of experiments, analysis, and ablations (**Reviewer-3WTy** describes our ablation studies as constituting a “detailed analysis of the method” whilst **Reviewer-WFbU** describes our hyperparameter exploration as “vast”). Despite this, we agree with the reviewer that extra ablation studies are always valuable for a deeper understanding of the method, and in order to address the comments:
> >
> >      > How well does the model work just with SVD alone? What is the influence of the number of iterations T?
> >
> >     we have accordingly provided a range of even more ablation studies of the objective function involving the requested value of T and the HOSVD in Fig 27. We also now visualise the learnt parts as a function of T, shown in Fig 28.
> >
> >     **Layer choice**
> >     > How does the layer choice matter? Does the user need to select the layer for decomposition or is there an automated way to choose?
> >
> >     It is well-known in the literature [6] that the earlier layers of StyleGAN affect more abstract high-level semantics whilst the later layers affect more fine-grained features like colours. We find this to be the case also when editing the feature maps, and this high/low-level separation of control largely appears across all convolutional generator architectures studied. One must indeed select the appropriate layer at which one wants to perform the decomposition, but we note that this manual step is a very common requirement for SOTA methods in the literature [7, 8, 9] working in the so-called “W+ space”.
> >
> >     **How editing works**
> >     > It is not explained how the editing works. Does the user inspect all discovered parts and then edit their spatial or semantic components? How does changing semantics work? Where does the changed appearance vector come from?
> >
> >     Editing is explained formally at the bottom of Section 3.3: the user selects the desired appearance from the columns of $\mathbf{A}$ (as is standard procedure in the unsupervised methods that learn directions in latent space [7, 10]) and desired part from $\mathbf{P}$, then adds the rank-1 tensor formed through the outer product of the two factors directly to the feature maps. Please also see our response (2) to Reviewer-3WTy, where we explain that the benefits of our unsupervised approach greatly outweigh the downsides. In particular, our unsupervised approach does not have the restrictive requirement of needing labels in the form of semantic masks to train the model. Annotating new datasets on which to train segmentation models to learn supervised models for new domains is thus far more labour-intensive than selecting desired columns from our factor matices. (Also see the relevant ease of manual selection discussed in Sect. C.7.2 of the manuscript).
> >
> > All remaining minor comments are addressed below:
> >
> > - `Other forms of clustering could be compared, e.g. k-means`: With the caveat that we do not just perform clustering (rather, a tensor factorisation to jointly model parts and appearances, where a clustering interpretation for the parts subproblem emerges from the specific design of the decomposition), one notable benefit in the formulation of our parts factors subproblem over k-means is the way in which it learns *soft* masks (rather than “hard” assignments). For example, our “eye” factor as seen in Fig. 11 focuses the variation on the eyeball and tapers off outside the centre according to the intensity of the semantics found at the relevant part, which helps blend the edit smoothly into the surrounding region when used. In contrast, a hard mask would put an equal amount of the appearance vector across the entire part, and thus is likely to result in edits that look more blocky and less natural.
> > - `do certain architectures lead to feature maps that are easier to decompose?`: Whilst we don’t notice a particular difference in decomposition difficulty (there isn’t a notable difference in the number of iterations needed for different generators for example), we do find the latent feature maps of StyleGAN2 to be particularly semantically rich relative to other architectures, which mirrors recent findings in the literature [11] demonstrating StyleGAN2’s more disentangled latent representations.
> > - `Are there substantial differences between different GANs trained on the same data`: There are not many examples of different generators pre-trained on the same datasets. That said, for StyleGAN2 and StyleGAN3 both trained on FFHQ we find the factors learnt in the early layers of StyleGAN2 to be more interpretable than those of StyleGAN3 (which again mirrors findings general findings from [11]).
> > - `Can appearance vectors of the same part be swapped between instances?`: Whilst this is not a focus of the paper, one can consider swapping the channel fibers under the learnt target parts, using the method of [12] to smoothly blend the two feature maps together.

---

> > > ### Author Response · Authors · 2022-11-17
> > > **Response to Reviewer-zd35 [references]**
> > >
> > >
> > >
> > > ----------
> > >
> > > * [1]: Zhu, Jiapeng et al. “Region-Based Semantic Factorization in GANs.” ICML (2022).
> > > * [2]: Melas-Kyriazi, Luke et al. “Finding an Unsupervised Image Segmenter in Each of Your Deep Generative Models.” ICLR (2022).
> > > * [3]: E. Collins, R. Bala, B. Price, and S. Süsstrunk, ‘Editing in Style: Uncovering the Local Semantics of GANs’, CVPR (2020).
> > > * [4]: C. Ding, X. He, and H. D. Simon, ‘On the Equivalence of Nonnegative Matrix Factorization and Spectral Clustering’, in Proceedings of the 2005 SIAM International Conference on Data Mining (2005).
> > > * [5]: Z. Yang and E. Oja, ‘Linear and Nonlinear Projective Nonnegative Matrix Factorization’, TNNLS (2010).
> > > * [6]: T. Karras, S. Laine, and T. Aila, ‘A Style-Based Generator Architecture for Generative Adversarial Networks’, CVPR (2019).
> > > * [7]: Shen, Yujun and Bolei Zhou. “Closed-Form Factorization of Latent Semantics in GANs.”, CVPR (2021).
> > > * [8]: Patashnik, Or et al. “StyleCLIP: Text-Driven Manipulation of StyleGAN Imagery.”, ICCV (2021).
> > > * [9]: Xu, Yinghao et al. “Generative Hierarchical Features from Synthesizing Images.”, CVPR (2021).
> > > * [10]: Härkönen, Erik et al. “GANSpace: Discovering Interpretable GAN Controls.” NeurIPS (2020).
> > > * [11]: Alaluf, Yuval et al. “Third Time's the Charm? Image and Video Editing with StyleGAN3.” *ArXiv,* (2022).
> > > * [12]: Suzuki, Ryohei et al. “Spatially Controllable Image Synthesis with Internal Representation Collaging.” *arXiv*  (2018).

---

> > > > ### Comment · Reviewer_zd35 · 2022-11-28
> > > > **Rebuttal response**
> > > >
> > > > Thank you for the clarifications and additional experiments. My questions and concerns have been answered and adressed well. Additionally, I did not find any strong concerns in the other reviews that were not adressed in the rebuttal. The updated paper incorporates all important points from the rebuttal and I recommend acceptance.

---

### Official Review · Reviewer_3WTy · 2022-10-24

**Confidence:** 4
**Correctness:** 4
**Technical Novelty And Significance:** 2
**Empirical Novelty And Significance:** 3
**Recommendation:** 8

**Clarity, Quality, Novelty And Reproducibility:**

## Clarity

The paper is quite clear and easy to follow.

## Quality

The results achieved by the model are quite good however some limitation do exist, see weakness (4) and some manual hyperparameter tuning is required (2)

## Novelty

As mentioned in weakness (1) I think the paper proposes a novel formulation but not a completely novel idea.

## Reproducibility

The paper should be reproducible.


**Strength And Weaknesses:**

## Strengths

+ The proposed method does not make any assumption on the architecture of the GAN (except for the generator to be convolutional). As such it can be applied to a vast array of existing models without requiring modifications.

+ The completely unsupervised nature of the formulation, together with the fast convergence time makes it very flexible and easy to be applied to many different generators and datasets (the authors tried it on 5 generators trained on 5 different datasets with good results).

## Weaknesses

1. The proposed method is remarkably similar to previous works in the literature (Collins 2020), with the main difference being the method used to cluster activations of the neural network. While Collins used k-Means, the authors proposed to use matrix factorization to identify patterns in the feature space. From my perspective the two approaches achieve a very similar outcome with a slightly different formulation. The authors of this work however explore the application of this clustering discovery more deeply than what was done in [Collins 2020]. I don’t agree with the point raised by the authors at the end of the related work section where they say that “Collins is limited to part swapping”, while their method is not. From my perspective their method is also doing part swapping but allows the extra flexibility of either swapping features between two images or using one of the learnt appearance factors.

2. The proposed method is completely unsupervised and task agnostic and as such can be applied out of the box to many different models and datasets. However the formulation does come with some hyperparameter that drastically can change the outcome: mostly the number of factors to extract and the layer of the generator to factorize. The ablation study shows that these two components, as expected, have a big impact on the factorization extracted and as such will need to be retuned for a new task and generator. Moreover once the factorization has been computed there’s still the need of some “human supervision” to map the learned factors to semantic concepts. As such this method might in the end require more human labor for practical application than competing works relying on pre-trained classifiers.

3. The proposed formulation implicitly assumes that the images are aligned while factorizing activations to compute the activations factors. While the authors show that the part factors can be aligned to a specific image with an ad hoc optimization, this is something done only in a second phase but not during the main optimization. From my perspective there’s no reason (except the computation cost) to not model separate part factors for each one of the activation maps used during the main factorization process. This will make the method more robust to datasets which are not very well aligned. With the current formulation the method cannot explain features that are not very well aligned across different samples.
A question I have for the authors is if there is anything preventing the use of a per sample part factorization during the main optimization?

4. The fact that the method performs very localized edits compared with the competitors can be seen both as a pro and as a cons. In particular it might be a limitation in the sense that it cannot adapt the full image to the particular edit being introduced. A practical example of this could be removing an object from an image and replacing it with background, but not removing the shadow casted by it. Some discussion about the limitations of the method along this direction will make the paper stronger.


**Summary Of The Paper:**

This work introduces a novel way of performing localized edits in the latent space of convolutional GANs like. The authors propose first to identify visual concepts by factorizing network activations at a certain layer into a set of appearance factors applied at certain spatial locations encoded by part factors. Once the factorization has been computed the learned appearance factors act as templates of visual concepts and can be used to identify parts of the generated image corresponding to a high level semantic concepts like “background” or “nose”. By swapping features in the latent space according to the localization provided by the appearance factors it is possible to apply local edits to images. The method is fast, architecture agnostic and by design limits the edits to only a user specified small portion of the image.

**Summary Of The Review:**

The idea behind the work is not novel, but the formulation is and the results are quite convincing. The method has some limitations but overall I feel like the authors made a good job at proving their contribution experimentally and providing a detailed analysis of the method through ablation studies. The main weakness of the paper is that it makes some strong assumptions in the factorization of the activations (weakness b). Lifting these would make the case for a stronger submission.

---

> ### Author Response · Authors · 2022-11-17
> **Response to Reviewer-3WTy [1/2]**
>
> We thank Reviewer-3WTy for the thorough review. We address each of the comments raised below:
>
>
> 1. As stated in the paper, we agree that part of our method bears some similarity to [1] in that our parts factor **subproblem** (considered in isolation) has interpretations as a form of soft-clustering [2, 3] of the feature maps' channel fibers. However, this is as far as both the methodological and practical similarities go. There are **major differences** between the two works both methodologically and in terms of goals and application that we argue Reviewer-3WTy does not acknowledge. Most importantly, we respectfully disagree that both approaches ”achieve a very similar outcome":
>
>     - **[1] does not learn factors for the** ***shared appearances and styles*** present throughout a dataset of images. This is a crucial difference in our work. We do not just perform clustering--we learn a *joint* decomposition of factors for parts, appearances, and their interactions, through a tensor factorisation.
>     - As such, the **majority of applications and tasks addressed in our paper are not possible with the method of [1]:**
>
>         In particular, one cannot use the method of Collins to, for example, remove objects (or add other generic, shared appearance concepts) within an image at arbitrary *precise* target spatial locations as we show in Fig. 3, nor can one localize the learnt appearance concepts in the images for visualization and interpretability of the semantics in generic convolutional generators as we demonstrate in Sect 4.1. In contrast to our method, [1] learns only something analogous to "parts" factors $\mathbf{P}$, not also "appearance" factors $\mathbf{A}$, which facilitates these many useful applications and insights (e.g. Fig. 14’s new background segmentation results).
>
>
>     In addition, **[1] is a StyleGAN-specific method, limited to swapping parts between two images**. Furthermore, they devise a relatively involved, multi-stage process involving the StyleGAN *latent code*, which requires further optimisation and tuning. **Thus, there is no clear way to extend their method to generic convolutional generators**. This is in contrast to our approach to learning coordinate-aware representations capable of performing editing directly and solely on the feature maps.
>
>
>   Following the reviewer's helpful comments we have updated the related work discussion to reflect more precisely how we employ the term "part-swapping": that is, between a reference and a source image (as opposed to our methodology that facilitates reference-image–free editing using the appearance concepts directly in feature map space). We hope this reply better elucidates the crucial differences, and we are more than happy to answer any follow-up questions.
>
>
> 2. We agree with Reviewer-3WTy that the proposed method inherits the drawbacks of all unsupervised methods in that manual supervision is needed to make the desired edits (as required in all related work [4, 5, 6, 7]). However, we argue the pros of our approach greatly outweigh the cons. For example, SOTA-supervised approaches (such as Wu et al. [8]) are restrictively dependent on semantic segmentation networks to detect locally-active style channels. This renders such supervised methods inapplicable for datasets for which there are no semantic networks available, whilst our unsupervised method has no such heavy constraints. Manually selecting the desired parts and appearances from a small set is orders of magnitude faster than annotating images so as to train supervised classifiers for new domains.
>
>     > The ablation study shows that these two components, as expected, have a big impact on the factorization extracted and as such will need to be retuned for a new task and generator.
>
>
>     Whilst $R_S$ indeed needs to be chosen, we see (e.g. from Fig. 24 in the appendix of the revised paper) that the parts factors are relatively robust to the choice of $R_S$ between 10 to 20, and we find common semantic regions such as the "eyes", "mouth", or "nose" across many different configurations of layers and values. Please find additional new runtime ablation studies in Fig. 28 in the appendix, where we see that regardless of the value of $R_S$, we consistently find parts that are semantically meaningful.
>
> **[continued below]**

---

> > ### Author Response · Authors · 2022-11-17
> > **Response to Reviewer-3WTy [2/2]**
> >
> > 3. There is nothing in principle preventing one from performing a sample-specific refinement in the main objective. However, as we discuss in response (2) to Reviewer-ixJz above, we find the extra complexity to the model form and the memory requirements not to be worth the sacrifice of the simplicity and desirable properties induced by the original objective. This optional two-step optimisation successfully tackles the wealth of tasks we address in the paper (and as a full methodology, *does* account for samples that are not necessarily aligned), and front-loading this additional complexity when not always necessary seems undesirable (e.g. the SOTA results presented in Tab. 2 use only the global parts factors). What’s more, the choice to learn just the global parts factors in the original objective allows one to scale up the decomposition to learn from the diversity present in many thousands of samples' feature maps--this becomes very expensive if one were to perform per-sample optimisation in the same step simultaneously.
> >
> > 4. We highlight that *strict* local editing is particularly challenging in GANs and sufficiently precise control is not possible in prior work—the recent NeurIPS’21 SOTA [5] states that their method’s inability to precisely control local objects in the way we can is the primary “shortcoming” of their work (for example, opening just a single eye in a photo, like is shown possible in Fig. 11 of our submission), whilst [6] from ICML’22 also lists this as a “common limitation”. We design our method to facilitate exactly this precise control currently lacking in the literature. That said, we thank the reviewer for putting forward a concrete example of a limitation that comes with such precision. We have added an extra paragraph to the manuscript to discuss such limitations. The revised submission now reads:
> >
> >     > Such strictly local editing means that after modifying a precise image region, any expected influence on the rest of the image is not automatically accounted for. As a concrete example, one can remove trees from an image, but any shadow they may have cast elsewhere is not also removed automatically.
> >
> > ----------
> >
> > * [1] E. Collins, R. Bala, B. Price, and S. Süsstrunk, ‘Editing in Style: Uncovering the Local Semantics of GANs’, CVPR (2020).
> > * [2]: Ding C, Li T, Jordan MI. ‘Convex and semi-nonnegative matrix factorizations’. IEEE Trans Pattern Anal Mach Intell. (2010).
> > * [3]: Z. Yang and E. Oja, ‘Linear and Nonlinear Projective Nonnegative Matrix Factorization’, TNNLS (2010).
> > * [4]: Shen, Yujun and Bolei Zhou. “Closed-Form Factorization of Latent Semantics in GANs.”, CVPR (2021).
> > * [5]: Härkönen, Erik et al. “GANSpace: Discovering Interpretable GAN Controls.” NeurIPS (2020).
> > * [6]: Zhu, Jiapeng et al. “Low-Rank Subspaces in GANs.” *NeurIPS* (2021).
> > * [7]: Zhu, Jiapeng et al. “Region-Based Semantic Factorization in GANs.” *ICML* (2022).
> > * [8]: Z. Wu, D. Lischinski, and E. Shechtman, ‘StyleSpace Analysis: Disentangled Controls for StyleGAN Image Generation’, CVPR (2021).

---

> > > ### Comment · Reviewer_3WTy · 2022-11-29
> > > **Rebuttal Response**
> > >
> > > Thank you for the clarifications.
> > > My questions and concerns have been answered and I have a more clear picture of the paper.
> > > I do not have strong concerns about this work and would suggest acceptance in its current form.
> > >
> > > I would also like to apologize to the authors for the grammatical mistakes in my original review since it was written in a rush.

---

### Official Review · Reviewer_ixJz · 2022-10-25

**Confidence:** 4
**Clarity, Quality, Novelty And Reproducibility:** The paper is clear and should be easi…
**Correctness:** 4
**Technical Novelty And Significance:** 3
**Empirical Novelty And Significance:** 3
**Recommendation:** 8

**Strength And Weaknesses:**

This is a very interesting paper that follows recent work on analyzing what happens in the feature maps of GANs. The idea of decomposing the feature map using an additive process is very sound. The results are also very good: parts are clear and the editing are OK.

That being said, there are some questions about this work:
1. The process is not entirely additive: the coefficient in $\Lambda$ can be negative, which means that some appearance and/or some spatial locations can be removed. Why not just making this non-negative as well? Sure it renders the optimization problem more complex, but it also corresponds more to the intuition given as motivation.
2. $\Lambda$ is arbitrarily set as $A^\top Z_i P$. Why? Why not optimize $\Lambda_i$ for each image? In classical NMF, instance coefficient are usually optimized along the atoms in the dictionary. And since you perform a few step of instance specific optimization for $P$ anyway, why not optimize it also.
3. In the editing, a rank 1 element is added to the original feature map. Why not removal as well? Why not reducing the weights of the appearance vectors that where at the spatial location as well?
4. Global spatial decomposition is problematic because it assumes a fixed layout, hence the instance specific optimization. Did you try a translation invariant formulation by, e.g., using a convolution operator in the decomposition? (this is very successful in NMF)
5. In 4.1.1, it is stated "frequently learns an appearance vector for a high-level background concept". Only one? So a single vector encodes all the possible backgrounds of the generator? That sounds implausible.

**Summary Of The Paper:**

This paper proposes a factorization of features maps in GANs such that spatial location is disentangled from appearance. Using NMF, the methods obtains a set of appearance prototypes and a set of non-negative spatial activations which, when combined with instance specific coefficients, reconstruct the feature map. Results show that this obtains localized compact regions corresponding to parts which are uncovered without supervision. These parts can be used to perform unsupervised object detection or even local editing by change the appearance prototype that is used at that location.

**Summary Of The Review:**

The idea is very sound and has practical application. There are a few question as to why the extra mile was not made in order to have a much more intuitive method.


-----

The rebuttal discuss most of my concerns which were more remarks and discussion material than actual requests for change.

---

> ### Author Response · Authors · 2022-11-17
> **Response to Reviewer-ixJz [1/2]**
>
> We thank Reviewer-ixJz for the assessment of our submission as a "very interesting paper" with "sound" methodological aspects. We address all comments in detail below:
>
>
> 1.
>     * Non-negative coefficients are an interesting suggestion and would certainly make sense in many other settings. However, the use of mixed-sign coefficients in this context is important in retaining the intuitive interpretation of the parts subproblem (complementary to that of our formulation as a particular tensor decomposition) as pursuing a form of soft-clustering, following the **semi**-NMF literature (e.g., Sect. 2.1 of [1]). In particular, it’s important that the *parts* factors are non-negative (the “cluster assignment matrix”), but that our coefficients (“centroids”) are of mixed-sign, so as to be able to reconstruct the **mixed-sign feature maps** of most GANs.
>
>     * More importantly, however, **learning the appearance factors with** **mixed-sign coefficients leads to a very intuitive editing interface**: a negative coefficient with the same appearance factor often affects a “semantically opposite” change to the image. For one concrete example, we find adding one particular appearance vector with a negative coefficient makes eyes “glance right” whilst a positive coefficient modifies the eyes to instead “glance left” in StyleGAN2 trained on FFHQ. For a different appearance factor, one can reduce the eyes’ size with a negative coefficient, as well as enlarge them with a positive one. Intuitively, it’s rather convenient to have this additional control of the opposite semantic in the same appearance factor.
>
> 2. As we detail in Sect. 3.3., $\Lambda_i=\mathbf{A}^\top\mathbf{Z}_i\mathbf{P}$ is not an arbitrary choice. This particular design of the objective function has two purposes: (i) to encourage both factor matrices to learn orthonormal columns (incentivising not only non-overlapping parts factors but also appearance factors that control "independent" concepts. For example, see the success of this form in encouraging orthogonality in the objective of [2]), and (ii) by writing the coefficients as a function of sample $i$'s feature maps, one can easily scale the shared decomposition to many thousands of feature maps as desired, without needing sample-specific coordinate optimisation for each $\mathbf{Z}_i\in\mathbb{R}^{C\times S}$ in the dataset. We highlight that, in contrast to traditional NMF-style approaches involving a single matrix $\mathbf{Z}_i\in\mathbb{R}^{C \times S}$, we are jointly factoring a collection of matrices stacked along the first mode of a tensor, $\mathcal{Z}\in\mathbb{R}^{N\times C \times S}$, learning global factor matrices shared between all $N$ samples.
>
>     > “And since you perform a few step of instance specific optimization for P anyway, why not optimize it also.”
>
>
>     As shown in the paper, these shared factors can already be used very successfully for en-masse local image editing (e.g. the global parts factors $\mathbf{P}$ give the SOTA results in Tab. 2) and interpretation of the generators' semantics through visualisation (Fig. 4). For this reason, we leave the refinement step as an optional choice when necessary or desired for specific samples or datasets.
>
>
> 3. Our formulation already facilitates the "removal" of appearance vectors at desired parts: in particular, we note that at the top of page 6, $\alpha\in\mathbb{R}$ is not constrained to be non-negative (which controls the "magnitude" of the edit). A negative value of $\alpha$ can be used just as readily and be seen to reduce the presence of the corresponding appearance vector at the desired part, by the desired amount. We encourage the reviewer to observe in the submitted supplementary material’s “compare.ipynb” notebook that a negative coefficient is used for the “smile” edit, for example.
>
> **[continued below]**

---

> ### Author Response · Authors · 2022-11-17
> **Response to Reviewer-ixJz [2/2]**
>
> 4. The idea of a translation-invariant formulation for the parts subproblem is indeed an interesting suggestion from Reviewer-ixJz worth exploring in the future.  However, such an extension is not as trivial as in the case of NMF since our formulation involves a tensor factorisation comprised of two subproblems for the parts and appearances’ factor matrices; the coefficients of which are written in a very particular functional form as described in (2) above. We would like to share with the reviewer some initial concerns:  in particular, Convolutional-NMF [3] breaks the orthogonality-preserving form of our model in Eq. 3 of our paper. Multiple shift-specific dictionaries $\mathbf{W}\_t$ are learnt in [3] which are decoupled from the coefficients $\mathbf{H}_{t\rightarrow}$ (which in our case are the parts factors $\mathbf{P}^\top$). However, as described above in response (2), the dictionary $\mathbf{W}$ (considering just the parts subproblem in isolation) is written as $\mathbf{Z}_i\mathbf{P}$ in our model to encourage sparse, non-overlapping parts factors through the reconstruction objective. Multiple, parts-independent dictionaries break the form that instils these important properties in the parts factors.
>
> 5. We agree that a single appearance vector that can add context-aware background over any image for a given generator and dataset is an interesting finding. However, this is more intuitive when one considers, for example, the remarkable properties of the StyleGAN generator, where the early layers are known to encode very high-level, abstract concepts (see Sect. 3 of the StyleGAN paper [4]), where this appearance vector is often found by our method (working in deep feature space, rather than pixel-space).
>
>
>     Evidence to support this claim can be seen in Fig 4. and for many samples in Fig. 18. This single appearance vector is consistently and highly active (in the elements of $\mathbf{a}_j^\top\mathbf{Z}_i$) in the “background” region across all image samples. We welcome the reviewer to observe this for themselves empirically should they wish, by playing with the notebook “localize_concepts.ipynb” provided in the supplementary material .zip, which demonstrates this finding. Finally, please note that we also now show more extensive “background segmentation” results using a single appearance vector for each of the 4 datasets studied to attest to this, in Fig. 14 of the revised paper's appendix.
> ----------
>
> * [1]: Ding C, Li T, Jordan MI. ‘Convex and semi-nonnegative matrix factorizations’. IEEE Trans Pattern Anal Mach Intell. (2010).
> * [2]: Q. V. Le, A. Karpenko, J. Ngiam, and A. Y. Ng, ‘ICA with Reconstruction Cost for Efﬁcient Overcomplete Feature Learning’, NeurIPS (2011).
> * [3]: Smaragdis, Paris. “Non-negative Matrix Factor Deconvolution; Extraction of Multiple Sound Sources from Monophonic Inputs.” *ICA* (2004).
> * [4]: T. Karras, S. Laine, and T. Aila, ‘A Style-Based Generator Architecture for Generative Adversarial Networks’, CVPR (2019).

---

### Official Review · Reviewer_WFbU · 2022-10-25

**Confidence:** 3
**Correctness:** 4
**Technical Novelty And Significance:** 3
**Empirical Novelty And Significance:** 3
**Recommendation:** 6

**Clarity, Quality, Novelty And Reproducibility:**

Though the method would not be applicable for models trained on general scene images like MS-COCO, the proposed method is clearly stated and novel enough, and its empirical results are impressive.

**Strength And Weaknesses:**

Strengths
- Dissecting a pre-trained generative model is an important and interesting task, and proposed method is novel and strong enough both in terms of its formulation and the quality of manipulation results.
- The goal of strict localized editing is well demonstrated qualitatively, and also quantitative evaluated with an appropriate evaluation metric.
- The authors present experimental results, and the changing trend in the decomposed features according to the hyperparameters’ change vastly in the supplementary material.

Weaknesses
- This method would only work for the models that are trained on images of single object type.


**Summary Of The Paper:**

The authors propose a way to manipulate intermediate latent features of pre-trained generative models to edit local parts of the image.
Specifically, they decompose a tensor of sampled latent features into two tensors, in which one could be interpreted as the set of appearance features and the other as the set of saliency maps that represent how much a specific appearance feature spans over the pixel-level space (coined as parts feature).
This is induced by constraining the decomposition procedure to make the parts feature nonnegative.
The decomposed features enable the manipulation of an original image by adjusting the attribute feature for the specific parts feature of interest.
As suggested by the results, the proposed method can be applied well to various pre-trained generative models with different architectures and training datasets.


**Summary Of The Review:**

This paper tackles an important problem which is helpful for both interpreting and further utilizing pre-trained generative models.
The proposed method is clear and novel, and its efficacy is well proved with various experimental results.

---

> ### Author Response · Authors · 2022-11-17
> **Response to Reviewer-WFbU**
>
> We thank Reviewer-WFbU for highlighting the novelty of the proposed method in addition to the strength of the experimental results and ablations provided. We agree that the application of the method is certainly more challenging for `"models trained on general scene images like MS-COCO"`. However, whilst we believe there does not exist a GAN trained on MS-COCO [1], we highlight that we already show in the paper results on BigGAN (a dataset frequently studied in related works [2, 3]) trained on the entirety of ImageNET (for example, Figs. 1b and 1c of the paper). As stated in [1]: `"In contrast to the popular ImageNet dataset, COCO has fewer categories but more instances per category."`, suggesting that ImageNET is likely to be equally as challenging as MS-COCO. We show that even in such a challenging setting, our method successfully locates the semantic parts of images for local image editing and that the appearance vectors allow one to localize semantically meaningful concepts across complex scenes (see Figs. 4c, 4d for examples of sea landscapes, or Fig 15. with churches).
>
> ----------
>
> * [1]: Lin, Tsung-Yi et al. “Microsoft COCO: Common Objects in Context.” ECCV (2014).
> * [2]: Shen, Yujun and Bolei Zhou. “Closed-Form Factorization of Latent Semantics in GANs.”, CVPR (2021).
> * [3]: Härkönen, Erik et al. “GANSpace: Discovering Interpretable GAN Controls.” NeurIPS (2020).

---

### Author Response · Authors · 2022-11-17
**Response to all reviewers**

We thank the reviewers for their thoughtful, detailed responses and positive comments:

- **Reviewer-WFbU** highlights the clarity and novelty of our method, the “impressive” results, and the importance of the task addressed.
- **Reviewer-ixJz** summarises our idea as “very sound [with] practical application” and as a “very interesting paper”.
- **Reviewer-3WTy** praises the clarity of the paper, “detailed analysis of the method”, and the “good job at proving [the] contribution experimentally”.
- **Reviewer-zd35** highlights the generality and speed of the method, and the evaluation on a “range of GAN architectures and datasets”.

We have addressed all questions in our initial responses, and we encourage all reviewers to also observe the **new** qualitative and quantitative **results** requested by Reviewer-zd35 in **Fig. 14**, in addition to the **extra page** of additional **ablation studies** in the supplementary material. We believe such results further validate the method and showcase additional utility of the learnt factors. Revisions to the revised paper are highlighted in red font. We thank all reviewers for their helpful suggestions, and we believe the revised paper is now much stronger as a consequence.

---

### Decision · Program_Chairs · 2023-01-20

**Decision:**

Accept: poster

**Justification For Why Not Higher Score:**


The method presented is similar to previous approaches e.g., Collins et al. 2020 about clustering feature activations of generative networks.

**Justification For Why Not Lower Score:**


All reviewers agree on the good empirical performance of the method and its simplicity.

**Metareview: Summary, Strengths And Weaknesses:**

The paper proposes a way to manipulate intermediate latent features of pre-trained generative models to edit local parts of the image. Specifically, they decompose a tensor of sampled latent features into two tensors, in which one could be interpreted as the set of appearance features and the other as the set of saliency maps that represent how much a specific appearance feature spans over the pixel-level space. This is induced by constraining the decomposition procedure to make the parts feature nonnegative. The decomposed features enable the manipulation of an original image by adjusting the attribute feature for the specific parts feature of interest. The proposed method can be applied well to various pre-trained generative models with different architectures and training datasets.
All reviewers agree on the good empirical performance of the method and its simplicity. The rebuttal submitted by the authors clarified the relation of the present work to Collins 2020, as well as its applicability beyond single object image.


**Note From Pc:**

if the above contains the word "oral" or "spotlight" please see: "oral" presentation means -> notable-top-5% and "spotlight" means -> notable-top-25%. As stated in our emails, we are disassociating presentation type from AC recommendations

**Summary Of Ac-Reviewer Meeting:**

N/A